# Why is Summertime Arctic Sea Ice Drift Speed Projected to Decrease?

Jamie L. Ward[1,2] and Neil F. Tandon[1]

[1]Department of Earth and Space Science and Engineering, York University, Toronto, ON, Canada
[2]Current affiliation: Cooperative Institute for Great Lakes Research, Ann Arbor, MI, USA

**Correspondence:** Neil F. Tandon (tandon@yorku.ca)

**Abstract.** Alongside declining Arctic sea ice cover during the satellite era, there have also been positive trends in sea ice Arctic-average drift speed (AADS) during both winter and summer. This increasing sea ice motion is an important consideration for marine transportation as well as a potential feedback on the rate of sea ice area decline. Earlier studies have shown that nearly all modern global climate models (GCMs) produce positive March (winter) AADS trends for both the historical period
and future warming scenarios. However, most GCMs do not produce positive September (summer) AADS trends during the historical period, and nearly all GCMs project decreases in September AADS with future warming. This study seeks to understand the mechanisms driving these projected summertime AADS decreases using output from 17 models participating in the Coupled Model Intercomparison Project phase 6 (CMIP6) along with 10 runs of the Community Earth System Model version 2 Large Ensemble (CESM2-LE). The CESM2-LE analysis reveals that the projected summertime AADS decreases are due to
changes in sea surface height (SSH) and wind stress which act to reduce sea ice motion in the Beaufort Gyre and Transpolar Drift. During March, changes in internal stress and wind stress counteract tilt force changes and produce positive drift speed trends. The simulated wintertime mechanisms are supported by earlier observational studies, which gives confidence that the mechanisms driving summertime projections are likely also at work in the real world. However, the precise strength of these mechanisms are likely not realistic during summer, and additional research is needed to assess whether the simulated summer-
time internal stress changes are too weak compared to changes in other forces. The projected summertime wind stress changes are associated with reduced sea level pressure north of Greenland, which is expected with the northward shift of the jet streams. The projected summertime SSH changes are primarily due to freshening of the Arctic Ocean (i.e. halosteric expansion), with thermal expansion acting as a secondary contribution. The associated ocean circulation changes lead to additional piling up of water in the Russian shelf regions, which further reinforces the SSH increase. Analysis of CMIP6 output provides preliminary
evidence that some combination of wind stress and SSH changes are also responsible for projected AADS decreases in other models, but more work is needed to assess mechanisms in more detail. Altogether, our results motivate additional studies to understand the roles of SSH and wind stress in driving changes of Arctic sea ice motion.

# 1 Introduction

One of the strongest signals of global climate change is the rapid retreat and thinning of Arctic sea ice, which strongly influences the global energy balance and atmosphere-ocean energy exchange. Long-term sea ice extent (SIE) changes are greatest during the summer melt season (Serreze and Meier, 2019), declining by 12.7% $y^{-1}$ during September 1979-2021, compared to a 2.6% $y^{-1}$ decline during March 1979-2021 (Meier et al., 2022). Record-low SIE has been frequently observed since the mid-2010s, with no record-high SIE observed during this time period (Parkinson and DiGirolamo, 2021; Meier et al., 2022). Furthermore, since 1958, Arctic-average end-of-summer sea ice thickness has declined by 66%, resulting in less multi-year sea ice cover (Kwok, 2018) and making existing ice more susceptible to melt. These trends exhibit large spatial variability, with sea ice concentration (SIC) decreasing most in the Barents, Kara, and Beaufort Seas (Comiso et al., 2017) and sea ice thickness (SIT) declining most rapidly in thick, multi-year ice regions in the central Arctic and along the northern coasts of Greenland and Canada (Bitz and Roe, 2004; Kwok, 2018).

Accurately simulating historical sea ice conditions using global climate models (GCMs) is crucial for revealing mechanisms responsible for sea ice evolution and building confidence in future sea ice projections. GCM performance assessment has been greatly facilitated by the successive phases of the Coupled Model Intercomparison Model (CMIP) alongside large ensembles of simulations using particular models like the Community Earth System Model version 1 Large Ensemble (CESM1-LE; Kay et al., 2015). Numerous studies comparing Arctic sea ice observations to CMIP and CESM1-LE output indicate that, although most GCMs can reproduce sea ice seasonality (e.g., Labe et al., 2018), they struggle to represent average sea ice conditions and trends (e.g., Kwok, 2011, 2018; Notz and Community, 2020; Keen et al., 2021; Watts et al., 2021). This is especially the case for SIT: only a handful of CMIP3 models reproduce observed winter SIT spatial patterns (Kwok, 2011), and SIT simulations only marginally improve for more recent CMIP phases (Stroeve et al., 2012; Watts et al., 2021).

Recent model developments, such as parameterizing surface melt ponds (Flocco et al., 2012), adding in "mushy layer" thermodynamics (Bailey et al., 2020) to represent sea ice surface properties more comprehensively, and reducing sea ice surface melt (Kay et al., 2022) produce output that better agrees with sea ice observations. However, given that state-of-the-art melt pond parameterizations still result in overestimated summer melt pond presence (e.g., Webster et al., 2022), and given our evolving understanding of sea ice thermodynamics, further model improvements are still needed.

In addition to thermodynamic melt and growth, sea ice motion can also induce sea ice change (e.g., Serreze and Meier, 2019; Wagner et al., 2021). Sea ice moves through the Arctic basin because of winds, ocean currents, ice internal stresses, momentum advection, the Coriolis force, and sea surface height (SSH) gradients, which are in turn coupled to SIT and SIC (e.g., Hibler, 1979; Connolley et al., 2004; Olason and Notz, 2014; Docquier et al., 2017; Spall, 2019). As ice floes move, they can diverge from or converge with existing ice (e.g., Kimura et al., 2013) or be exported through Arctic Ocean gates like Fram Strait (Kwok et al., 2009; Smedsrud et al., 2017).

The effects of sea ice motion on regional growth and melt processes vary seasonally and depend on existing ice pack properties (Chevallier and Salas-Mélia, 2012). In the summer, ice divergence enhances local melt as the low-albedo ocean surface absorbs more incoming sunlight. During the cold season when regional surface air temperatures are well below freezing,

sea ice leads allow for energy transfer from the ocean to the atmosphere to promote ice formation (Kimura et al., 2013). Converging sea ice enhances SIT, making it more susceptible to thinning (Flato and Hibler III, 1995; Bitz and Roe, 2004) but less likely to melt completely by the end of the summer (Chevallier and Salas-Mélia, 2012). In turn, these SIT changes influence sea ice deformation (e.g. Docquier et al., 2017) and the roles of wind, ocean, and internal stresses on observed sea ice drift (e.g., Steele et al., 1997; Roach and Blanchard-Wrigglesworth, 2022).

Arctic sea ice dynamical changes can be characterized in terms of changes in Arctic-average drift speed (AADS). Observed AADS has increased over the past 40 years in tandem with sea ice loss, especially in the summer months (Rampal et al., 2009; Tandon et al., 2018; Zhang et al., 2022). However, GCMs have varying success in reproducing these trends. Rampal et al. (2011) computed annual sea ice drift speed trends using output from CMIP3 models and found that these GCMs significantly underestimate observed AADS increase. However, Tandon et al. (2018) found that much of this disagreement stems from analyzing model and observational data with different temporal resolutions. Nonetheless, CMIP5 models still underestimate observed summertime AADS increase over 1979-2014 (Tandon et al., 2018) for reasons that are unclear.

Tandon et al. (2018) also found a strong seasonal contrast in projected AADS trends under the Representative Concentration Pathway 8.5 (RCP8.5) scenario, under which emission changes through the end of the 21st century produce a globally averaged top-of-atmosphere radiative forcing of 8.5 W m$^{-2}$. Under this scenario, March AADS steadily increases until the late 21st century for most CMIP5 GCMs, while September AADS trends switch from positive to negative in the early- to mid-21st century. These projected summertime decreases in Arctic sea ice motion are the focus of this study because they sharply contrast with expectations based on historical trends.

Understanding the mechanisms of these projected summertime trends is an important step toward assessing the realism of models and improving confidence in their projections. To this end, we analyze sea ice dynamics-related output from the Coupled Model Intercomparison Project phase 6 (CMIP6) with additional in-depth analysis of the Community Earth System Model version 2 Large Ensemble (CESM2-LE). This analysis reveals that changes in SSH and wind stress likely drive these projected decreases in summertime AADS. In section 2, we provide details regarding the model output and our analysis methods. In section 3, we discuss CMIP6 and CESM2-LE trends of AADS, and we perform a detailed breakdown of the CESM2-LE sea ice momentum budget and SSH changes. Section 4 provides further discussion and concluding remarks.

## 2   Data and Methods

In this study, we examine output from 17 models participating in the Coupled Model Intercomparison Project, Phase 6 (CMIP6) (Eyring et al., 2016), as listed in Table 1. For all models, we examine Arctic sea ice drift speed and related quantities for March (the month of maximum SIE) and September (the month of minimum SIE) over the period 1950-2100. We use the historical simulations of CMIP6 for the period 1950-2014 and the Shared Socioeconomic Pathway 585 (SSP585) scenario for the period 2015-2100. SSP585 is considered a "business-as-usual" scenario with strong warming, similar to the CMIP5 RCP8.5 scenario (O'Neill et al., 2017), which was analyzed by Tandon et al. (2018). We chose this scenario because it maximizes the climate change signal compared to the noise generated by internal climate variability. We analyze only one simulation from each

**Table 1.** Models and year ranges examined in this study. Except for CESM2-LE, all model simulations were performed as part of CMIP6. See section 3.1 for additional details regarding how the periods of increasing and decreasing September sea ice Arctic-average drift speed (AADS) trends were determined. The information for CESM2-LE is based on the CESM2-LE ensemble average.

| Model (Ensemble) | March Years | September Years (Positive AADS Trends) | September Years (Negative AADS Trends) |
|---|---|---|---|
| ACCESS-CM2 | 1950-2100 | 1950-2027 | 2028-2100 |
| AWI-CM-1-1-MR | 1950-2100 | 1950-1992 | 1993-2100 |
| BCC-CSM2-MR | 1950-2100 | 1950-1990 | 1991-2100 |
| CESM2 (CMIP6) | 1950-2100 | 1950-2015 | 2016-2100 |
| CESM2 (CESM2-LE) | 1950-2100 | 1950-2017 | 2018-2100 |
| CESM2-WACCM | 1950-2100 | 1950-2022 | 2023-2100 |
| CNRM-CM6-1 | 1950-2100 | 1950-2011 | 2012-2100 |
| CNRM-ESM2-1 | 1950-2100 | 1950-2010 | 2011-2100 |
| EC-Earth3-CC | 1950-2093 | 1950-2014 | 2015-2067 |
| IPSL-CM6A-LR | 1950-2083 | 1950-2008 | 2009-2059 |
| MIROC6 | 1950-2100 | 1950-2058 | 2059-2100 |
| MIROC-ES2L | 1950-2100 | 1950-2060 | 2061-2084 |
| MPI-ESM1-2-HR | 1950-2100 | 1950-2000 | 2001-2072 |
| MPI-ESM1-2-LR | 1950-2100 | 1950-1987 | 1988-2074 |
| MRI-ESM2-0 | 1950-2100 | 1950-2009 | 2010-2100 |
| NESM3 | 1950-2100 | 1950-2011 | 2012-2100 |
| NorESM2-LM | 1950-2100 | 1950-2040 | 2041-2100 |
| NorESM2-MM | 1950-2100 | 1950-2100 | – |

CMIP6 model, using the r1i1p1f1 variant label whenever possible to maintain consistent sampling and forcing across different models. Here, the numbers next to "r," "i," "p" and "f" provide labels for the ensemble member, initialization method, physics package, and forcing datasets, respectively. For MIROC6, MIROC-ES2L, CNRM-CM6-1 and CNRM-ESM2-1, the r1i1p1f1 variant was not available, and we instead use the r1i1p1f2 variant, which uses an updated version of the external forcing (Durack and Taylor, 2022). For CESM2, only the r4i1p1f1 variant was available, which is identical to r1i1p1f1 except for a small perturbation applied to the initial state.

Following Tandon et al. (2018), we calculate sea ice drift speed from daily output of drift velocity components (CMIP6 variable names "siu" and "siv"). As discussed in Tandon et al. (2018), this time resolution is needed when comparing sea ice drift speed in models to drift speed computed from daily buoy observations. For computing spatial averages over the Arctic, we use the same domain as used in Tandon et al. (2018): we include all grid points north of 79°N for longitudes 124°W eastward to 103°E, and we include all grid points north of 68°N over all other longitudes. This domain essentially includes all Arctic sea ice except within the Barents and Kara Seas (where there is little to no sea ice cover during September) and the Canadian

Arctic Archipelago. Because we wish to focus on drifting (not landfast) sea ice, we also exclude grid points within 150 km of Arctic coastlines, as in previous studies (Rampal et al., 2011; Tandon et al., 2018). We compute the AADS by calculating the drift speed at each latitude-longitude grid point and then taking the area-weighted average over the Arctic domain. (Grid box area has CMIP6 variable name "areacello.")

In the model output files, drift velocity in regions without sea ice are assigned a special "missing value" that indicates that the data are missing. For calculations of monthly averages, we exclude any days on which sea ice is missing, and we exclude any months for which all values in that month are missing. When computing monthly-average AADS, any points where sea ice is missing for all days in that month are excluded from the spatial average. For trend calculations, we exclude any points where the number of monthly-mean sea ice drift velocity samples is less than five. We have also tested performing trend calculations with a minimum sample size of three months, and our results did not show strong sensitivity to that choice. For all trend calculations (including trends of quantities other than sea ice drift velocity), we mask out regions where sea ice drift velocity trends have not been computed.

Because of the coordinate singularity at the North Pole on spherical polar grids, sea ice and ocean models commonly use displaced-pole and tripolar grids with the pole(s) placed over land. When calculating sea ice drift speed, we use the drift velocity components on the model's native grid. That is, drift speed equals $\sqrt{u_n^2 + v_n^2}$, where $u_n$ and $v_n$ are the velocity components on the native grid. However, when examining the velocity components separately, we transform the velocity components to align with the eastward and northward directions using the relationships

$$
\begin{aligned}
u &= u_n \cos\theta - v_n \sin\theta, \\
v &= u_n \sin\theta + v_n \cos\theta,
\end{aligned}
\tag{1}
$$

where $\theta$ is the computed angle between the $x$ direction on the native grid and the $x$ (eastward) direction on a spherical polar grid, and $u$ and $v$ are the (spherical polar) eastward and northward drift velocity components, respectively. This transformation is applied to the output for all CMIP6 models except BCC-CSM2-MR, whose output was already transformed to eastward and northward components (Xiaoyong Yu, personal communication, 2022).

To better understand the dynamical mechanisms responsible for Arctic sea ice drift speed changes, we analyze wind stress, ocean stress, internal stress and tilt forces. However, the CMIP6 models did not provide output of these quantities at daily resolution, which is needed in order to decompose terms in the momentum budget. For this reason, we also analyze 10 ensemble members of the Community Earth System Model version 2 Large Ensemble (CESM2-LE; Rodgers et al., 2021), which includes all of the daily output needed for our analysis. The CESM2-LE ensemble members were generated through small perturbations to the 1950 initial state. For the 1950-2014 period, CESM2-LE performs the same historical simulation as in CMIP6. Output from SSP585 (the scenario analyzed for CMIP6) was not available, so for CESM2-LE during 2015 onward, we instead analyze output from the SSP370 scenario. The amount of warming in SSP370 is less than in SSP585, but the reduced signal is not a concern because noise in CESM2-LE can be filtered out by averaging across the large ensemble. We discuss below additional details regarding the responses to the different scenarios.

For all of the CESM2-LE analysis presented in this study, we show ensemble-averaged results unless otherwise stated. The specific output variables used were "siu_d" and "siv_d" for the sea ice drift components, "uarea" for grid box area, "strairx_d" and "strairy_d" for the wind stress components, "strocnx_d" and "strocny_d" for the ocean stress components, and "strintx_d" and "strinty_d" for the sea ice internal stress components. We compute the sea surface tilt force by computing the spatial gradients of the SSH output (variable "SSH_2"). To obtain the sea ice mass per unit area, we multiply the daily sea ice thickness (variable "sithick_d") by the sea ice density ($917 \, \text{kg m}^{-3}$), which is constant in CESM2. All quantities (e.g. sea ice drift speed and individual terms in the sea ice momentum budget) are first computed at the daily timescale, and the results are averaged over each month, excluding days on which sea ice is missing. Ensemble averages are produced by averaging these monthly timeseries over all ensemble members.

## 3  Results

### 3.1  Arctic sea ice motion trends

Here, we characterize the Arctic-average and regional sea ice drift speed changes in CMIP6 and CESM2-LE output. In Fig. 1, we show March and September average AADS timeseries with a 21-year smoothing window applied. (See Section 2 for details. For comparison, unsmoothed AADS timeseries are shown in Fig. A1.) As was the case for CMIP5 (Tandon et al., 2018), there is a large (more than a factor of two) intermodel spread in AADS during the historical period. Tandon et al. (2018) argued that, in CMIP5, much of this spread could be explained by choices of prescribed model parameters, such as the air-ice drag coefficient. In CMIP6, March AADS increases over most of the 1950-2100 period (Fig. 1a).

During September, AADS trends are positive during most of the 20th century, but then they become negative for all but one model (NorESM2-MM) during the 21st century (Fig. 1c). The periods of increasing and decreasing September AADS trends are also indicated in Table 1. For each model, these periods are determined by first identifying the year during which the 21-year smoothed September AADS reaches a maximum value during the study period. The portion of the study period before and including this year is considered to be the period of positive September AADS trend, and the portion of the study period after this year is considered to be the period of negative September AADS trend. The year of this AADS trend sign change varies widely among the CMIP6 models, ranging from 1992 for AWI-CM-1-1-MR to 2058 for MIROC6, with a majority of the models (11 out of 17) placing this year during 2000-2030. Tandon et al. (2018) show qualitatively similar results from CMIP5, with increasing March AADS trends over most of 1950-2100, with almost all models producing a switch from positive to negative AADS trends during September.

Projected negative September trends in the 21st century vary greatly in magnitude, with some models depicting faster slow-down (e.g., AWI-CM-1-1-MR and CESM2-WACCM) compared to others (e.g., MRI-ESM2.0 and NorESM2-LM). In contrast with other models, IPSL-CM6A and EC-Earth3-CC produce March AADS decreases and September AADS increases near the end of the study period. Tandon et al. (2018) also found that some CMIP5 models from other modelling centres exhibit decreasing March AADS in the late 21st century. Buoy observations show increasing AADS trends during both March and

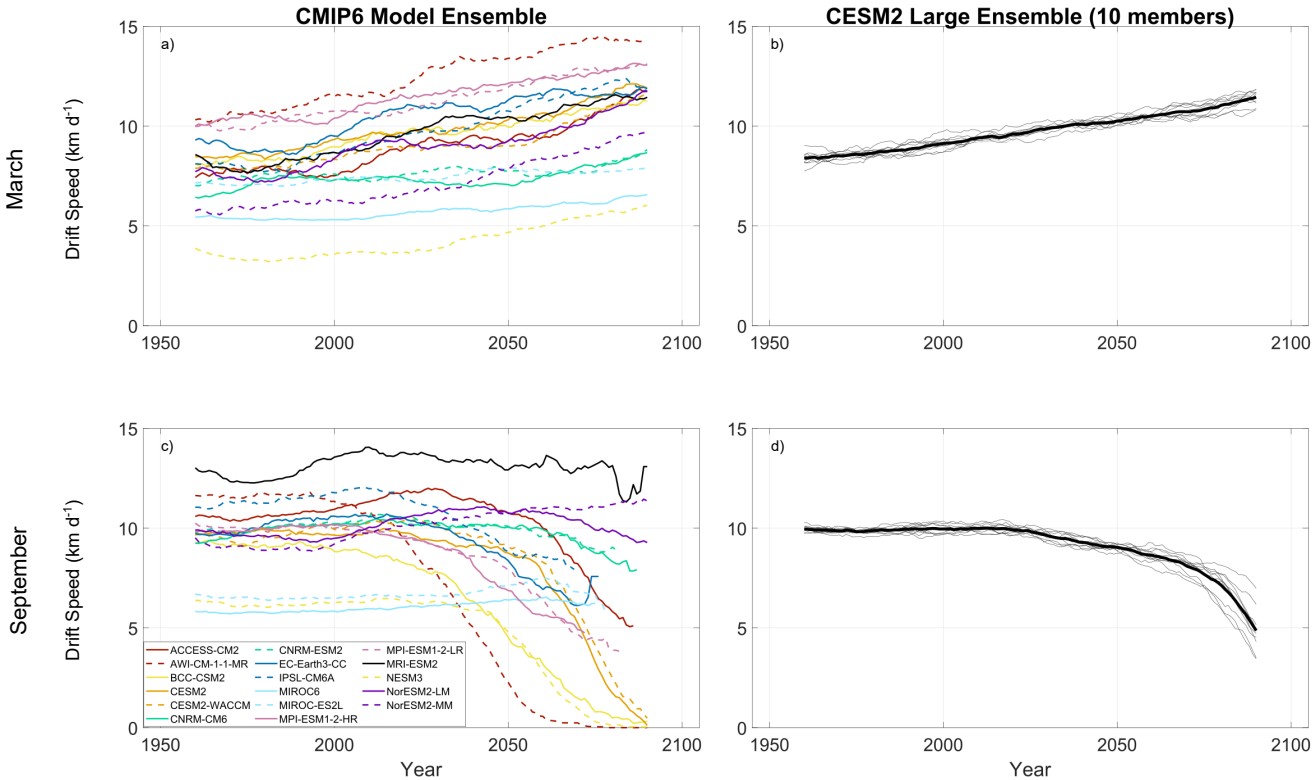

**Figure 1.** AADS simulated in (a,c) CMIP6 and (b,d) CESM2-LE during (a,b) March and (c,d) September. Each monthly mean timeseries of each CMIP6 model and each CESM2-LE ensemble member is smoothed with a 21-year moving window. In panels b and d, the gray lines indicate individual ensemble members and the thick black lines indicate ensemble averages.

September (Tandon et al., 2018). Thus, the decreases in September AADS projected by nearly all models contrasts dramatically with historical trends and is worthy of further investigation.

In CESM2-LE, March AADS increases almost linearly for the entire time series, starting at $\sim 8 \, \text{km d}^{-1}$ in 1960 and increasing to $\sim 11 \, \text{km d}^{-1}$ by 2090 (Fig. 1b). In contrast, September AADS remains close to $10 \, \text{km d}^{-1}$ between 1960 and 2018 before decreasing to $5 \, \text{km d}^{-1}$ at the end of the study period (Fig. 1d). The CESM2-LE ensemble spread during September also increases towards the end of the study period. Thus, in agreement with the CMIP6 models, CESM2-LE produces positive March AADS trends over the entire study period (Fig. 1b) and negative September AADS trends over most of the 21st century (Fig. 1d). The negative September trend is especially clear 2025 onward, and much of our analysis hereafter will focus on 2025-2100 trends.

As mentioned in Section 2, the CMIP6 simulations follow the SSP585 scenario while CESM2-LE follows the SSP370 scenario during 2015 onward. The single CESM2 simulation in CMIP6 (Fig. 1a,c) shows stronger AADS trends compared to CESM2-LE (Fig. 1b,d), which is expected since there is stronger warming in SSP585 compared to SSP370. Nonetheless, the overall results of CESM2-LE fall within the range produced by the CMIP6 models. Thus, we expect that our analysis of mechanisms in CESM2-LE will be relevant for understanding mechanisms in CMIP6 simulations under the SSP585 scenario.

Fig. 2 provides more regional detail by showing the March sea ice motion averaged over the end of the historical period (1991-2010, left column) and the end of the 21st century (2071-2090, right column). During both time periods, there is anticyclonic motion in the Beaufort Gyre, and the highest drift speeds are in the eastern Arctic Ocean along the Transpolar Drift (Fig. 2a,b), as found in observations (Kwok, 2011; Howell et al., 2016). Drift speeds are markedly larger in 2071-2090 than in 1991-2010 over the western Arctic, with little to no change in drift speeds along the Transpolar Drift. Thus, much of the simulated increase in March AADS appears to be due to increased drift speed in the Western Arctic. Tandon et al. (2018) attributed this increased drift speed primarily to reduced sea ice thickness, since the spatial correspondence between positive drift speed trends and negative sea ice thickness trends was apparent over the entire Arctic basin. In contrast, the spatial correspondence between drift speed and surface wind speed trends was less consistent and confined to relatively small regions. However, we will present analysis below indicating that wind stress changes play a stronger role than suggested by earlier work.

Examining the velocity components separately will facilitate the detailed investigation of the sea ice momentum budget that we present later in this study. Comparing the zonal velocity component during the historical and future periods (Fig. 2c,d), we see increasing eastward drift north of Greenland, decreasing westward drift north of the Barents Sea, and increasing westward drift in the Beaufort, Chukchi and East Siberian (BCES) Seas. The meridional velocity component (Fig. 2e) shows northward flow north of Russia and southward flow toward Fram Strait, indicative of the Transpolar Drift crossing the North Pole. There is weak meridional drift in the Beaufort Sea, where drift is mostly in the zonal direction. Comparing the meridional velocity component during the historical and future periods (Fig. 2e,f), we see overall very little change, with a very slight decrease in northward drift north of Russia. There is also a slight decrease in southward drift north of Canada and a slight increase in northward drift off the Alaskan coast. Altogether, this velocity component analysis indicates that much of the increase in March Arctic-average drift speed is due to increases in the zonal velocity magnitude in the western Arctic.

In contrast, September drift speeds decrease over most of the Arctic between the historical and future periods (Fig. 3). 1991-2010 drift speeds range from 8-10 km d$^{-1}$, and the velocity vectors show Transpolar Drift and Beaufort Gyre features qualitatively similar to but quantitatively greater than in March (compare Figs. 3a and 2a). The September 2071-2090 average depicts lower drift speeds throughout the Arctic, most dramatically in the BCES Seas, where drift speed drops from $\sim 7$ km d$^{-1}$ to nearly zero (Fig. 3a,b). This decrease combined with weaker decreases elsewhere in the Arctic results in a more spatially heterogeneous drift speed field during the future period compared to the historical period. The velocity components reveal that much of the speed reduction in the BCES Seas is due to reduced westward drift in these areas (Fig. 3c,d). Previous analysis of CMIP6 models indicates that these areas exhibit the earliest transition to seasonal sea ice cover, regardless of the particular emission scenario (e.g., Årthun et al., 2021). The more modest decreases in drift speed elsewhere in the Arctic are associated with a reduction in westward drift north of Canada and in the Transpolar Drift north of Fram Strait (Fig. 3c,d). The latter

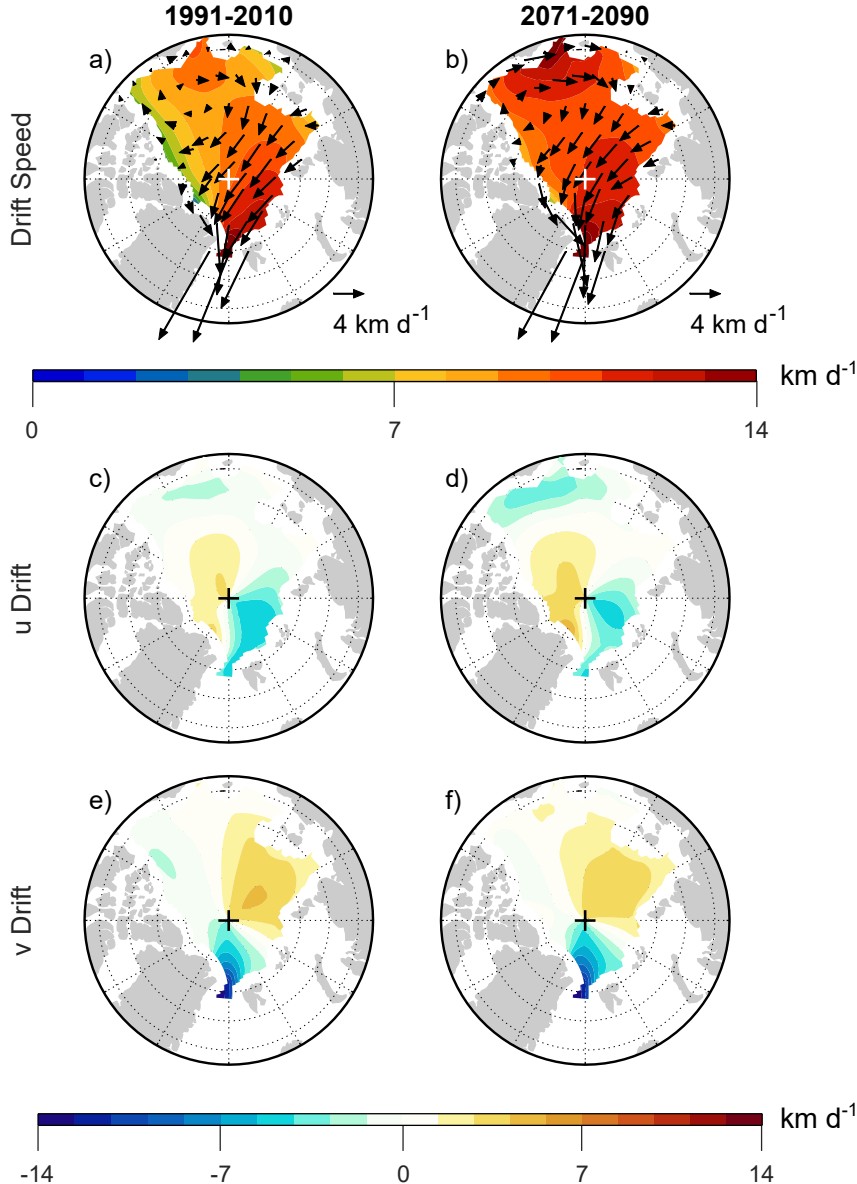

**Figure 2.** CESM2-LE March monthly mean sea ice (a,b) drift speed magnitude (shading) and drift vectors, (c,d) eastward drift component, and (e,f) northward drift component averaged over (a,c,e) 1991-2010 and (b,d,f) 2071-2090. Note that, because of the nonlinearity in computing drift speed, the shading values in panels (a,b) might differ from the lengths of the overlying vectors in regions where the drift direction is highly variable on submonthly timescales. For the eastward drift component (c,d), positive (negative) values indicate counterclockwise (clockwise) motion around the North Pole. For the northward drift component (e,f), positive (negative) values indicate motion toward (away from) the North Pole.

change has also been observed in early 21st century sea ice observations, and it has contributed to a reduction in sea ice volume export through Fram Strait (Spreen et al., 2020). Reduced drift speeds also arise from changes in the meridional drift component, namely a reduction in northward drift over most of the eastern Arctic and a reduction in southward drift over most of the western Arctic (Fig. 3d,e). As was the case in March, however, the largest changes in September drift speed appear to be associated with changes in the zonal drift component.

The changes highlighted in Figs. 2-3 are also apparent in the projected (2025-2100) regional trends, shown in Fig. 4 shading. During March, the strongest trends are positive with overall higher values in the western Arctic than in the eastern Arctic (Fig. 4a shading). These trends are associated primarily with increased eastward drift north of Canada and Greenland, accompanied by increased westward drift in the Beaufort Sea (Fig. 4a vectors and Fig. 4c shading). Based on differences between time periods, there appeared to be an increase in Beaufort Gyre circulation in the Chukchi and East Siberian Seas (Fig. 2), but the trends are not westward in these locations (Fig. 4a,c).

The western Arctic also shows positive trends in the meridional drift component (Fig. 4e shading), but this is a region where the climatology (contours) is weakly negative. Thus, the positive $v$ trend here mainly indicates decreasing southward drift, with possibly a transition toward slight northward drift, and such a change is not expected to contribute substantially to increased drift speed. There is also some increasing southward drift near Fram Strait (Fig. 4e), but this is accompanied by decreasing westward drift (Fig. 4c), and there is no noticeable effect on drift speed. North of Scandinavia, there is decreasing westward drift (Fig. 4c) accompanied by decreasing northward drift (Fig. 4e), contributing to slightly negative drift speed trends here (which is not noticeable on the shading scale in Fig. 4a). Altogether, these changes further establish that the positive trend in Arctic-average March drift speed is primarily due to increased zonal drift in the western Arctic, which was also apparent in Fig. 2.

During September, there are negative drift speed trends throughout the Arctic (Fig. 4b), and the changes in drift velocity indicate weakening of the Beaufort Gyre and Transpolar Drift (Fig. 4d,f). There is clear weakening of both the zonal and meridional velocity components throughout the Arctic. The strongest drift speed trends are in the BCES Seas, which appear to be primarily due to changes in the zonal drift component. These changes further establish that the negative trend in Arctic-average September drift speed is primarily due to reduced westward drift in BCES Seas, as was also apparent in Fig. 3, and weakening of northward and westward motion in the Transpolar Drift is also an important contribution. Interestingly, there are positive zonal drift trends north of Canada and Greenland, where there are also positive zonal drift trends during March (Fig. 4b,c). However, the climatology of the zonal drift component shows a strong seasonal contrast, with westward drift during September and eastward drift during March. As a result, the positive zonal drift trends act to reduce drift speed during September and increase it during March.

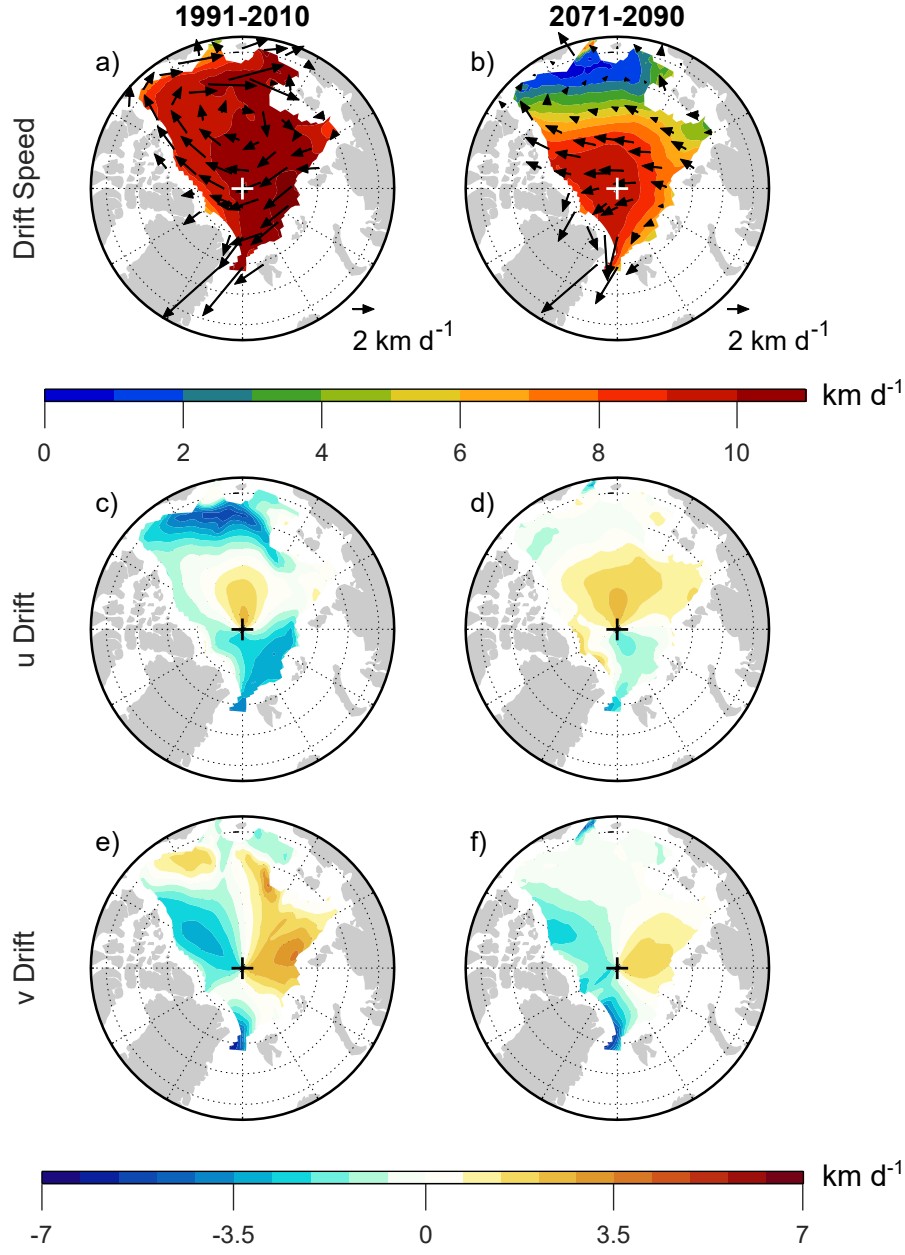

**Figure 3.** As in Fig. 2, but for September. Note that the shading and vector scales are different from those in Fig. 2.

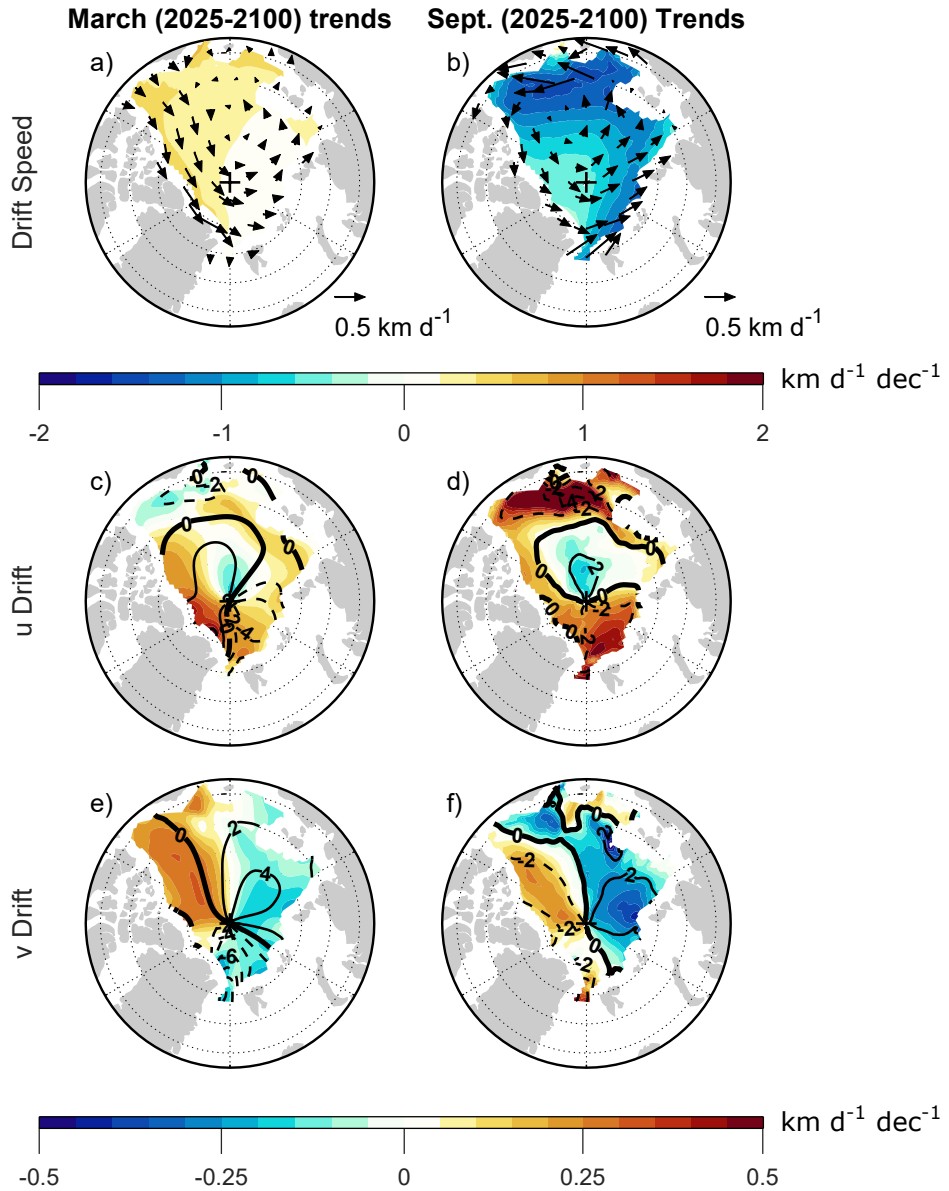

**Figure 4.** Shown in shading are the CESM2-LE trends computed over 2025-2100 during (a,c,e) March and (b,d,f) September for (a,b) sea ice drift speed (c,d) the eastward sea ice drift component, with positive (negative) values indicating anomalous counterclockwise (clockwise) motion around the North Pole, and (e,f) the northward sea ice drift component, with positive (negative) values indicating anomalous motion toward (away from) the North Pole. Vector velocity trends are overlaid in panels a and b. Trends are computed from monthly mean timeseries as detailed in section 2. Contours in panels c-f show climatological values averaged over 2015-2035, with contour interval of 2 km d$^{-1}$. Negative contours are dashed and the zero contours are thick.

## 3.2 Drivers of projected sea ice motion trends

What are the physical mechanisms driving these projected sea ice drift changes? To address this question, we begin with the sea ice momentum equation,

$$\frac{\partial \mathbf{v}}{\partial t} + \mathbf{v} \cdot \nabla \mathbf{v} + \mathbf{f} \times \mathbf{v} = -g\nabla H + \frac{1}{m_A}(\tau_{\mathbf{a}} + \tau_{\mathbf{w}} + \mathbf{F_i}), \tag{2}$$

where $\mathbf{v} = u\mathbf{i} + v\mathbf{j}$ is the drift velocity, $\mathbf{f} = f\mathbf{k}$ is the Coriolis parameter, $\tau_{\mathbf{a}} = \tau_{ax}\mathbf{i} + \tau_{ay}\mathbf{j}$ is the wind stress, $\tau_{\mathbf{w}} = \tau_{wx}\mathbf{i} + \tau_{wy}\mathbf{j}$ is the ocean stress, $\mathbf{F_i} = F_{ix}\mathbf{i} + F_{iy}\mathbf{j}$ is the force due to sea ice internal stress, $g = 9.8 \text{ m s}^{-2}$ is the gravitational acceleration, $H$ is the SSH, and $m_A$ is the sea ice mass per unit area (e.g., Hibler, 1979). Since we are interested in long-term (multidecadal) averages and trends, we can assume that velocity variations on this timescale are slow enough that the time tendency term ($\partial \mathbf{v}/\partial t$) can be neglected. We have also performed analysis verifying that, on the timescales of interest here, momentum advection [the second term on the left hand side (LHS)] is negligible, as it is two orders of magnitude smaller than other terms. Thus, over long timescales, the Coriolis term balances the other forces, and the velocity components are

$$u = -\frac{g}{f}\frac{\partial H}{\partial y} + \frac{1}{fm_A}(\tau_{ay} + \tau_{wy} + F_{iy}),$$
$$v = \frac{g}{f}\frac{\partial H}{\partial x} - \frac{1}{fm_A}(\tau_{ax} + \tau_{wx} + F_{ix}), \tag{3}$$

where subscripts $x$ and $y$ denote zonal and meridional components, respectively. To facilitate our discussion, we refer to the SSH-related tilt forces as "geostrophic," and we define the geostrophic velocity components as $u_g = -\frac{g}{f}\frac{\partial H}{\partial y}$ and $v_g = \frac{g}{f}\frac{\partial H}{\partial x}$. We refer to the remaining terms as "ageostrophic," with components $u_a = \frac{1}{fm_A}(\tau_{ay} + \tau_{wy} + F_{iy})$ and $v_a = -\frac{1}{fm_A}(\tau_{ax} + \tau_{wx} + F_{ix})$ (Armitage et al., 2017). Thus, each velocity component is equal to the sum of its geostrophic and ageostrophic components, i.e. $u = u_g + u_a$ and $v = v_g + v_a$. The geostrophic terms in (3) are linear, but as we discuss further below, the ageostrophic terms are potentially nonlinear because sea ice mass can vary on timescales that are not well separated from the timescales on which the sea ice stresses vary.

To test the validity of equation (3), we have computed ensemble mean trends of the force terms on the right hand side (RHS) of (3), and we have compared their sum, which we call the "reconstructed trend" (Fig. A2), to the trend of the total velocity field. (See section 2 for additional details regarding our temporal and ensemble averaging approach.) During March, the total velocity trends and the reconstructed trends are indistinguishable (Figs. 4c,e and A2a,c). During September, the total velocity trends and the reconstructed trends are qualitatively similar (Figs. 4d,f and A2b,d), except near Fram Strait, where the reconstructed meridional component trend is opposite in sign to the total meridional component trend (Figs. 4f and A2d). Quantitatively, however, the reconstructed September trends are about 50% smaller than the total velocity trends.

We have found that, when we attempt to reconstruct the change in velocity between two days from daily fields by temporally integrating equation (2)—i.e. without neglecting any terms in the momentum equations—the results are reasonably accurate during March but highly inaccurate during September (not shown). Thus, it appears that the quantitative difference between the reconstructed and total September velocity trends is not due to the approximations in (3) and is instead due to strong subdaily variability in sea ice motion during September compared to March. This subdaily variability would lead to inaccuracy when

attempting to reconstruct the monthly mean ageostrophic velocity from daily output, since the ageostrophic terms in (3) are nonlinear. (Subdaily variability would not impact the reconstruction of monthly mean geostrophic velocity, since the tilt force terms are linear.) Numerous earlier studies have shown substantial subdaily variability in Arctic sea ice motion that is present during most months, but nearly disappears during winter freeze-up (Hibler et al., 1974; McPhee, 1978; Colony and Thorndike, 1980; Heil and Hibler, 2002). Based on these earlier studies, it is to be expected that our reconstructed long-term trends are quantitatively more accurate during March than they are in September.

Because of the inaccuracy in reconstructing the ageostrophic velocity from individual force terms, we instead take the approach of computing the ageostrophic velocity as the difference between the total velocity and the geostrophic velocity, i.e. as the residuals $u_a = u - u_g$ and $v_a = v - v_g$. Fig. 5 shows the March and September trends of $u_a$ (computed as a residual) along with the trends of $u_g$. Here, we focus on trends of the zonal drift component, since as shown above, this component captures the key features relevant for trends in Arctic-average drift speed. Comparing the residual ageostrophic trend (Fig. 5c,d) with the ageostrophic trend reconstructed from individual force terms (Fig. A3), we see that the residual and reconstructed trends are indistinguishable during March (Figs. 5c and A3a), but the residual trend during September is quantitatively an order of magnitude larger than the reconstructed trend (Figs. 5d and A3b). These differences further confirm the influence of nonlinearity on the computation of ageostrophic velocity.

During March, there is an overall positive trend in $u_g$, with peak trends north of Russia and Greenland (Fig. 5a). In contrast, there are negative trends in $u_a$ throughout the BCES Seas and north of Scandinavia and Russia, with positive trends elsewhere (Fig. 5c). The negative $u_a$ trend in the BCES Seas is strong enough to produce a negative total $u$ trend in the Beaufort Sea (Fig. 4c). North of Scandinavia and Russia, the negative $u_a$ trends act to offset positive $u_g$ trends, resulting in weakly positive total $u$ trends here. Positive $u_a$ trends north of Canada and Greenland act to amplify the $u_g$ trends here, and the strongest total $u$ trends are in this region. This analysis indicates that March trends are due to a combination of geostrophic and ageostrophic trends. Most notably, both geostrophic and ageostrophic processes substantially contribute to the drift acceleration north of Canada and Greenland, which as noted above, is a key feature responsible for the positive trend in March AADS.

Intriguingly, the geostrophic trends during March and September are nearly identical (Fig. 5a,b), suggesting that processes other than sea ice melt are responsible for generating the sea surface tilt changes. (We will revisit this matter below when analyzing SSH changes.) Thus, the contrast between the March and September trends is mainly due to ageostrophic effects. In particular, the deceleration of total $u$ north of Alaska and in the central Arctic (Fig. 4d) is primarily due to ageostrophic changes (Fig. 5d), whereas the deceleration elsewhere is due to a combination of ageostrophic and geostrophic changes (Fig. 5b,d).

What processes are responsible for the ageostrophic velocity trends? To address this question, Fig. 6 shows trends for the internal stress, wind stress and ocean stress terms in the zonal momentum budget. During March, the negative trend in the Beaufort Sea is arising from changes in internal stress (Fig. 6a), whereas the positive trend north of Canada and Greenland is coming primarily from changes in wind stress (Fig. 6c). There is strong cancellation between wind stress and ocean stress trends (Fig. 6c,e), as expected since a change in wind stress is expected to generate an opposing frictional drag by the ocean surface (e.g., Figure 1 of Nakayama et al., 2012). We refer to the combination of wind and ocean stresses as "wind-ocean" stress (Fig. 6g). The wind-ocean stress trend is qualitatively opposite to the internal stress trend (Fig. 6a,g), indicating that

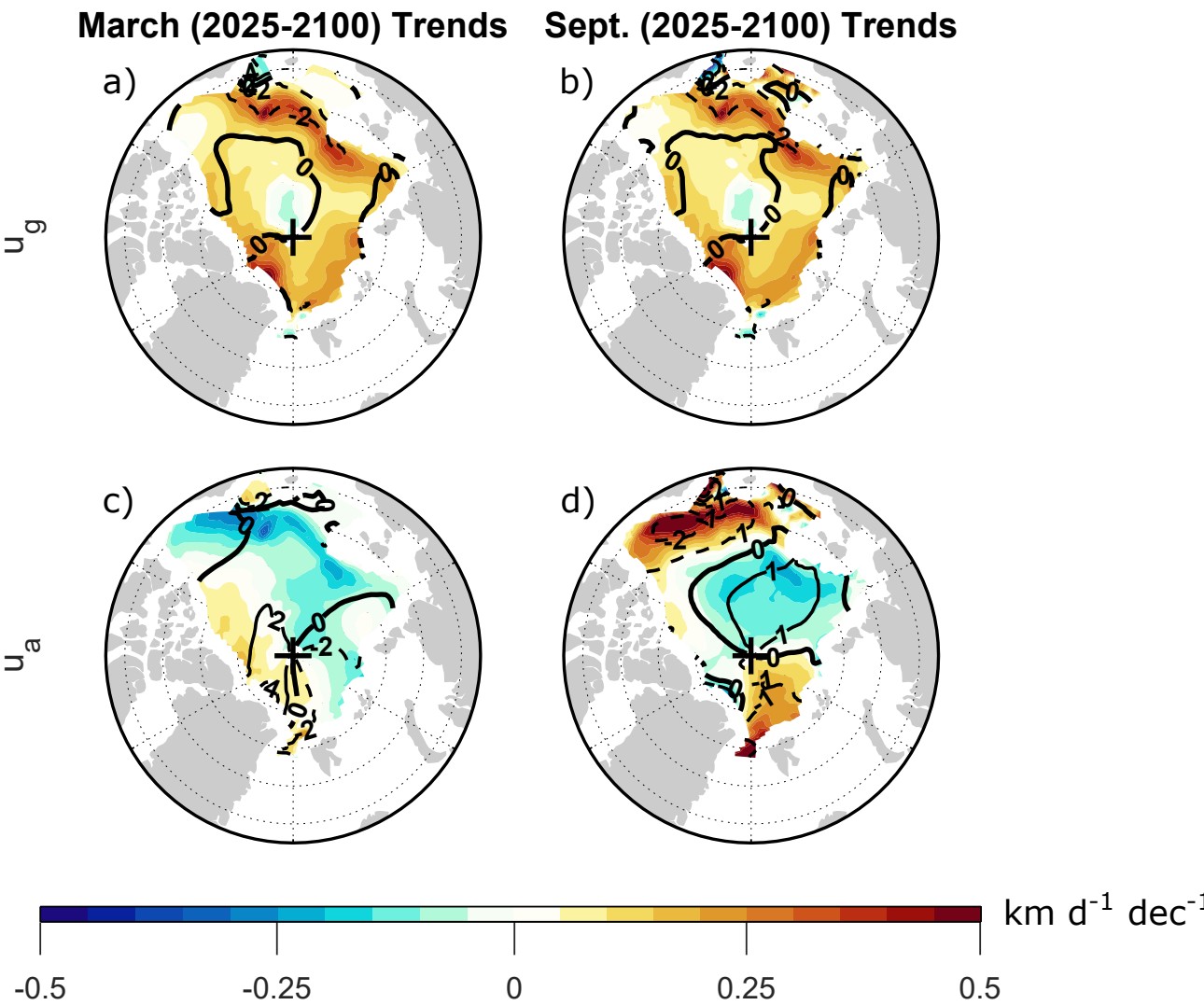

**Figure 5.** Shown in shading are the CESM2-LE trends computed over 2025-2100 for (a,b) the geostrophic zonal drift component, and (c,d) the ageostrophic zonal drift component during (a,c) March and (b,d) September. The ageostrophic velocity is computed as a residual as described in the text. Trends are computed from monthly mean timeseries as detailed in section 2. Contours show climatological values averaged over 2015-2035, with contour intervals of (a-c) 2 km d$^{-1}$ and (d) 1 km d$^{-1}$. Negative contours are dashed and the zero contours are thick.

the internal stress is reacting to the net force exerted by wind-ocean stress. If there were only normal stresses and no shear stresses in the sea ice, such a change in the internal stress could be understood much like a spring's internal force reacting in the opposite direction to an applied force. However, in reality and in models, there are also shear stresses in the sea ice and more work is needed to assess the possible contribution of shear stresses to the internal stress changes. Earlier observational studies

have also found regional variation in the dominance of internal stress versus wind stress on sea ice motion trends (e.g., Spreen et al., 2011), which provides some confidence in the overall realism of the CESM2-LE simulations. However, there is strong internal variability influencing the observed trends (Spreen et al., 2011) which confounds a more detailed regional comparison with the CESM2-LE trends.

As we stated above, we do not expect the computation of monthly mean ageostrophic terms from daily output to be quantita-
315 tively accurate during September. Nonetheless, we can qualitatively compare the ageostrophic velocity trends with the trends of individual force terms in order to assess which processes are likely contributing. During September, there are opposing trends of wind stress and ocean stress (Fig. 6d,f), and there are opposing trends of internal stress and wind-ocean stress (Fig. 6b,h). This opposition was also apparent in the March trends (Fig. 6a,c,e,g), although the spatial structures of the March trends are very different from those of the September trends. The September wind-ocean stress trends (Fig. 6h) qualitatively resemble
the September $u_a$ trends (Fig. 5d) over most of the Arctic, whereas there is little qualitative resemblance between the internal stress trends (Fig. 6b) and the $u_a$ trends. This analysis suggests that changes in wind-ocean stress are primarily responsible for the decreases in ageostrophic velocity during September. For sufficiently thin sea ice, such an outcome is expected, since the sea ice would approach a free drift state for which internal stress changes play a negligible role.

Moreover, the sign of the largest $u_a$ changes (in the BCES Seas and north of Fram Strait) is opposite to the ocean stress
changes (Fig. 6f) in these regions and matches the sign of the wind stress changes (Fig. 6d). This analysis suggests that changes in wind stress are likely the dominant driver of the ageostrophic deceleration during September, with ocean stress changes acting as a modulating influence. More rigorous quantitative attribution of mechanisms would require subdaily data which can hopefully become available sometime in the future. Such wind stress changes are expected with a reduction of sea level pressure (SLP) concentrated north of Greenland, which would act to slow down the Beaufort Gyre and Transpolar Drift.
Indeed, such an SLP reduction is evident in CESM2-LE (Fig. 7b) as well as in simulations with other models (e.g., Vavrus et al., 2012), but historical SLP trends are unclear because of strong internal variability (e.g., Deser and Teng, 2008). Such an SLP decrease is expected with the northward shift of the tropospheric jet streams projected by most climate models (e.g., Yin, 2005; Miller et al., 2006; Barnes and Polvani, 2013), which in turn brings greater frequency of cyclones over the western Arctic (Moore et al., 2018). Thus, we would expect other models to produce wind stress changes that contribute to negative
September AADS trends, although we would expect the amplitude of that wind stress contribution to vary considerably among models owing to strong intermodel variability in jet shift and SLP trends (e.g., Miller et al., 2006).

In CMIP6 models during September, we found some correspondence between SSH trends, SLP trends and sea ice velocity trends (not shown), which suggests that some combination of tilt force and wind stress changes is responsible for projected summertime AADS decreases in other models. However, more investigation of these models is needed to quantitatively assess
the roles of tilt force and wind stress changes compared to changes in other forces. As additional daily and subdaily model output becomes available for other models, we will be able to develop greater confidence in how we expect the contribution of wind stress changes to compare to the contribution of SSH changes. Despite this uncertainty, our analysis suggests that tilt force changes are an important contribution to projected summertime sea ice motion decreases, which motivates additional investigation of the reasons for the associated SSH changes.

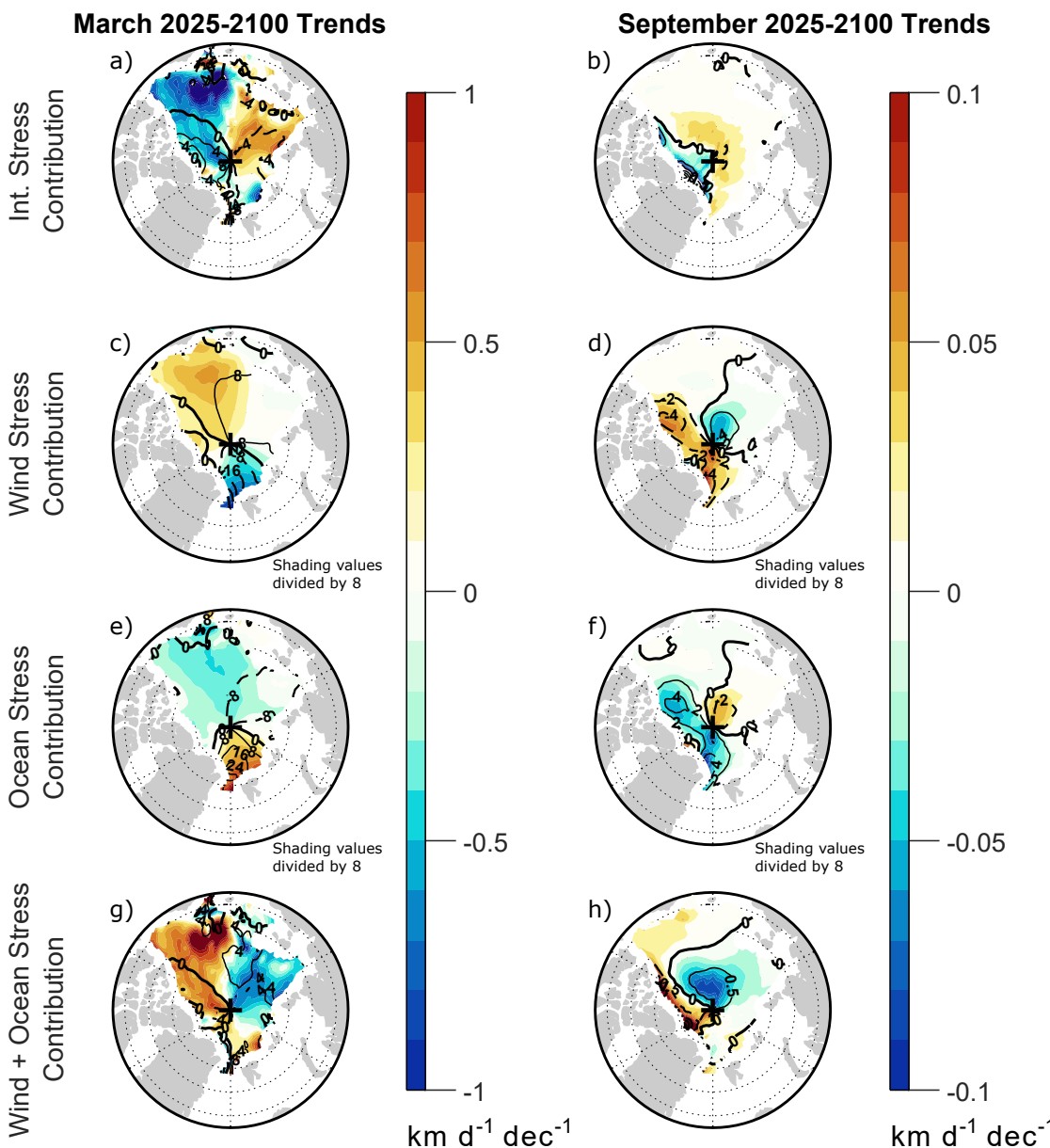

**Figure 6.** Shown in shading are the (a,b) internal stress, (c,d) wind stress, (e,f) ocean stress, and (g,h) wind plus ocean stress contributions to the 2025-2100 trends of the zonal component of sea ice drift velocity in CESM2-LE during (a,c,e,g) March and (b,d,f,h) September. Trends are computed from monthly mean timeseries as detailed in section 2. Contours show climatological values averaged over 2015-2035, with contour intervals of (a,g) 4 km d$^{-1}$, (b,h) 0.5 km d$^{-1}$, (c,e) 8 km d$^{-1}$, and (d,f) 2 km d$^{-1}$. Negative contours are dashed and the zero contours are thick. For clarity, the shading values in panels c-f have been divided by eight.

### 345   3.3   Drivers of projected summertime SSH trends

Our earlier analysis motivates further investigation of the projected SSH changes during September. Figure 7a, shading, shows that SSH is projected to increase over almost the entire Arctic, with the exception of a slight projected decrease in the central Arctic. The projected SSH increases are generally higher ($\sim 3 \ \mathrm{cm \ dec}^{-1}$) along the periphery of the basin than over the central Arctic. Such a pattern would act to flatten the Arctic "SSH dome" that has been well-documented in observations and 350 models (e.g., Koldunov et al., 2014). Fig. 7a shows that the weakened meridional SSH gradient corresponds with the anomalous cyclonic drift pattern (vectors), reiterating the contribution of SSH changes (in addition to wind stress changes) to the decreases in sea ice motion, as we established previously through analysis of the sea ice momentum budget.

Following Gill and Niller (1973), the contributions to a change in SSH, $\eta'$, can be expressed as

$$\eta' = -\frac{p_a'}{g\rho_0} - \frac{1}{\rho_0} \int_{-H}^{0} \rho' \, dz + \frac{p_b'}{g\rho_0}, \tag{4}$$

where $p_a'$ is the change in SLP exerted by the atmosphere, $\rho'$ is the change in water density, $\rho_0 = 1029 \ \mathrm{kg \ m}^{-3}$ is a reference value for water density, $p_b'$ is the change in bottom pressure, $z$ is vertical distance with negative values below the mean surface level, and $H$ is the ocean depth at a given location. The first, second and third terms on the RHS represent the contributions of surface pressure changes, steric density changes and bottom pressure changes, respectively. While the terms in (4) are arranged to solve for SSH change, this expression could also be rearranged into an expression solving for bottom pressure, in which 360 case it is equivalent to stating that a change in bottom pressure is determined by the combined effects of atmospheric pressure change, SSH change and density change within the ocean column. The only approximation entering into this expression is that the contribution of an SSH change to bottom pressure is assumed to be Boussinesq.

We now perform calculations from CESM2-LE output to assess each of these contributions. According to (4), if SLP changes were the dominant driver of SSH changes, then SLP and SSH changes should be of opposite sign. Indeed, SLP trends are 365 negative over most of the Arctic (Fig. 7b), but the peak negative trend lies north of Greenland rather than over the central Arctic. This result suggests that SLP trends are likely not the dominant driver of SSH trends. Equation (4) indicates that, if bottom pressure were the dominant factor, we would expect SSH and bottom pressure trends to be of the same sign. However, bottom pressure trends are negative over most of the basin (Fig. 7e), indicating that it cannot be the primary explanation for the positive SSH trends. These results suggest that the SSH trends are primarily due to steric changes.

To get an initial sense of the steric changes, Fig. 7c shows trends of sea surface temperature (SST), which, not surprisingly, is positive over the entire basin. Such warming on its own would be expected to produce decreased density and increased SSH. Sea surface salinity (SSS) trends (Fig. 7d) are negative north of Greenland and Russia. The spatial structure of these SSS trends are similar to those found in the annual mean for CMIP5 and CMIP6 models, and they show a strong correspondence with river runoff increases (Wang et al., 2022). These SSS decreases would act to reduce water density and contribute to increased SSH, 375 while the positive SSS trends elsewhere would act to reduce SSH. These spatially varying SSS trends might be combining with the more spatially uniform SST warming to produce the spatially varying SSH trends. These results provide additional evidence that SSH trends are due to steric changes.

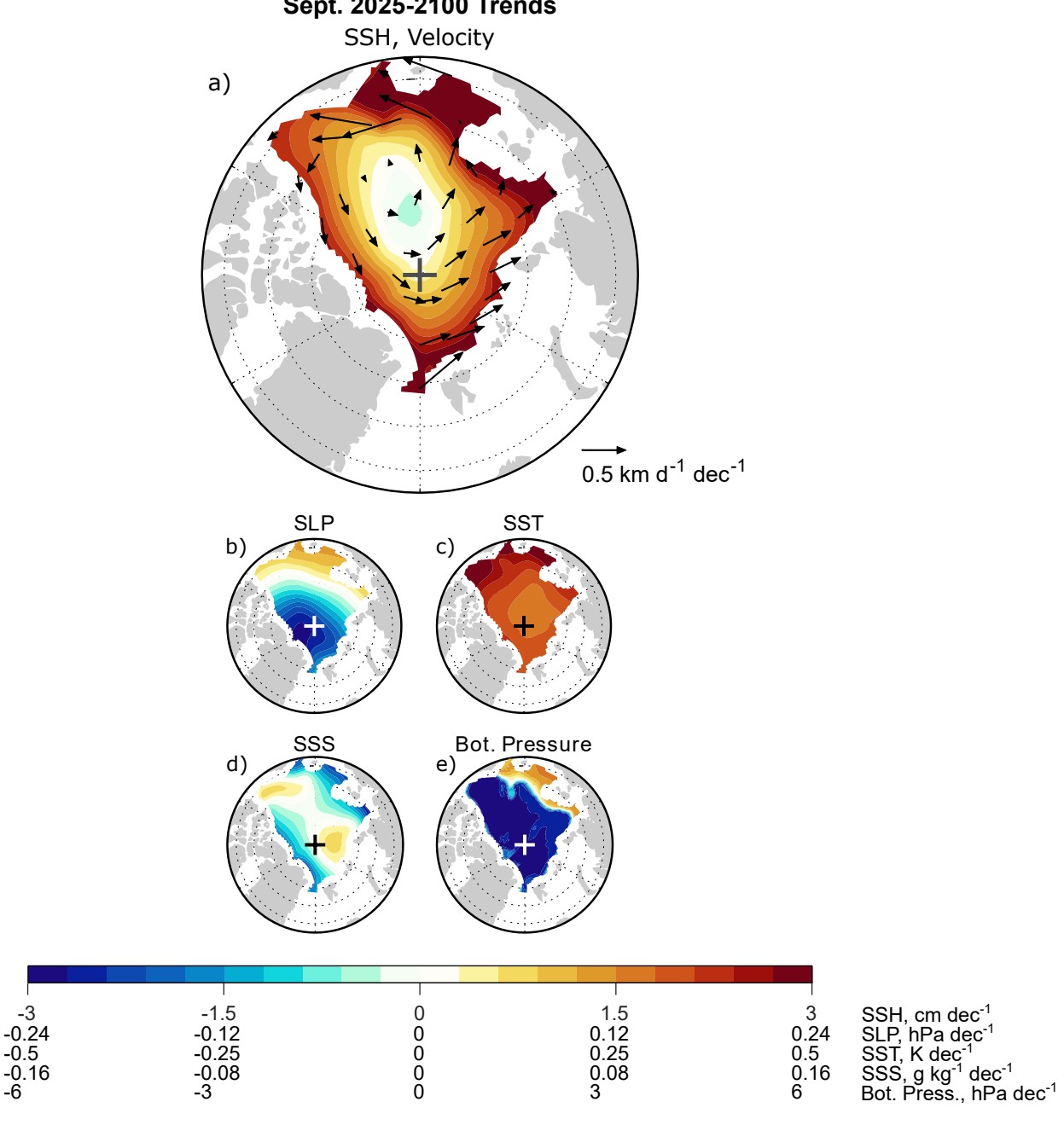

**Figure 7.** CESM2-LE September 2025-2100 trends of (a) SSH (shading) and drift velocity (vectors), (b) sea level pressure, (c) sea surface temperature, (d) sea surface salinity, and (e) ocean bottom pressure. Trends are computed from monthly mean timeseries as detailed in section 2.

However, to more rigorously attribute thermosteric and halosteric effects, the depth-integrated density (not just surface values) must be considered. To this end, we first define the steric SSH change, $\eta_s' \equiv -\frac{1}{\rho_0} \int_{-H}^{0} \rho' \, dz$ (i.e. the second term on the RHS of eq. 4). This quantity can then be expanded as

$$\eta_s' = \alpha \int_{-H}^{0} T' \, dz + \beta \int_{-H}^{0} S' \, dz, \tag{5}$$

where $T'$ is the temperature change, $\alpha$ is the thermal expansion coefficient, $S'$ is the salinity change and $\beta$ is the (negative) saline expansion coefficient (e.g., Gill and Niller, 1973; Rao and Tandon, 2021). On the RHS, the first term represents the thermosteric contribution to SSH change, and the second term represents the halosteric contribution. Although observed $\alpha$ and $\beta$ vary by depth (e.g., Fig. 2 in MacIntosh et al., 2017), we have chosen constant values $\alpha = 7.5 \times 10^{-5} \, \mathrm{K}^{-1}$ and $\beta = -7.6 \times 10^{-4} \, \mathrm{kg} \, \mathrm{g}^{-1}$ for all ocean grid points, since specifying $\alpha$ and $\beta$ at each level produces nearly identical results (not shown).

Using these values along with the monthly three-dimensional output of ocean salinity and temperature (CESM2 variables "SALT" and "insitu_temp," respectively), we have computed halosteric (Fig. 8a) and thermosteric (Fig. 8b) contributions to SSH trends. These results reveal that, while both the halosteric and thermosteric contributions to SSH trends are positive, the halosteric contribution to SSH trends is approximately 50% larger than the thermosteric contribution. This comparison is in agreement with expectations from earlier studies: although thermosteric effects dominate over halosteric effects over most of the global ocean, halosteric effects dominate in the Arctic Ocean because of the near-freezing water temperatures and the relatively low thermal expansion coefficient (Koldunov et al., 2014; Carret et al., 2017). Decadal freshwater variability in the Arctic Ocean has been the greatest contributor to observed SSH change (e.g., Xiao et al., 2020; Lyu et al., 2022), and freshwater changes have been shown to dominate projected SSH changes in CMIP6 under the SSP126 and SSP585 warming scenarios (e.g., Zanowski et al., 2021). In contrast to the contributions implied by SSS changes (Fig. 7d), the depth-integrated halosteric contribution in CESM2-LE is positive over the entire domain (Fig. 8b), suggesting that subsurface freshening of the Arctic Ocean is playing an important role.

The analysis of Li and Fedorov (2021) suggests that such subsurface freshening is due to advection of salt from the deep Arctic Ocean to lower latitudes in the Atlantic Ocean, and the weakened Atlantic Meridional Overturning Circulation results in greater accumulation of salt at lower latitudes compared to higher latitudes. However, observational analysis suggests that there has been increased penetration of Atlantic waters into the Eurasian basin, and most models (including CESM2) do not reproduce this "Atlantification" (Muilwijk et al., 2023). Depending on how Atlantification competes with surface freshening, there may be reductions rather than increases in SSH in the Eurasian Basin, which would affect changes in sea ice motion. Thus, further work is needed to improve the model representation of Atlantification in the Eurasian Basin in order to improve confidence in model projections of sea ice motion.

Simulated bottom pressure trends are negative over most of the domain (Fig. 8c), with positive trends over the Russian shelf regions. These positive trends appear to be due to anomalous ocean transport toward these shelves (as can be inferred from sea ice drift vector trends in Fig. 7a), which would be expected to pile up water and increase SSH in these areas. Over the interior Arctic basin, however, bottom pressure is more responsive to column-integrated density, which is decreasing. The bottom

## Sept. 2025-2100 Trends

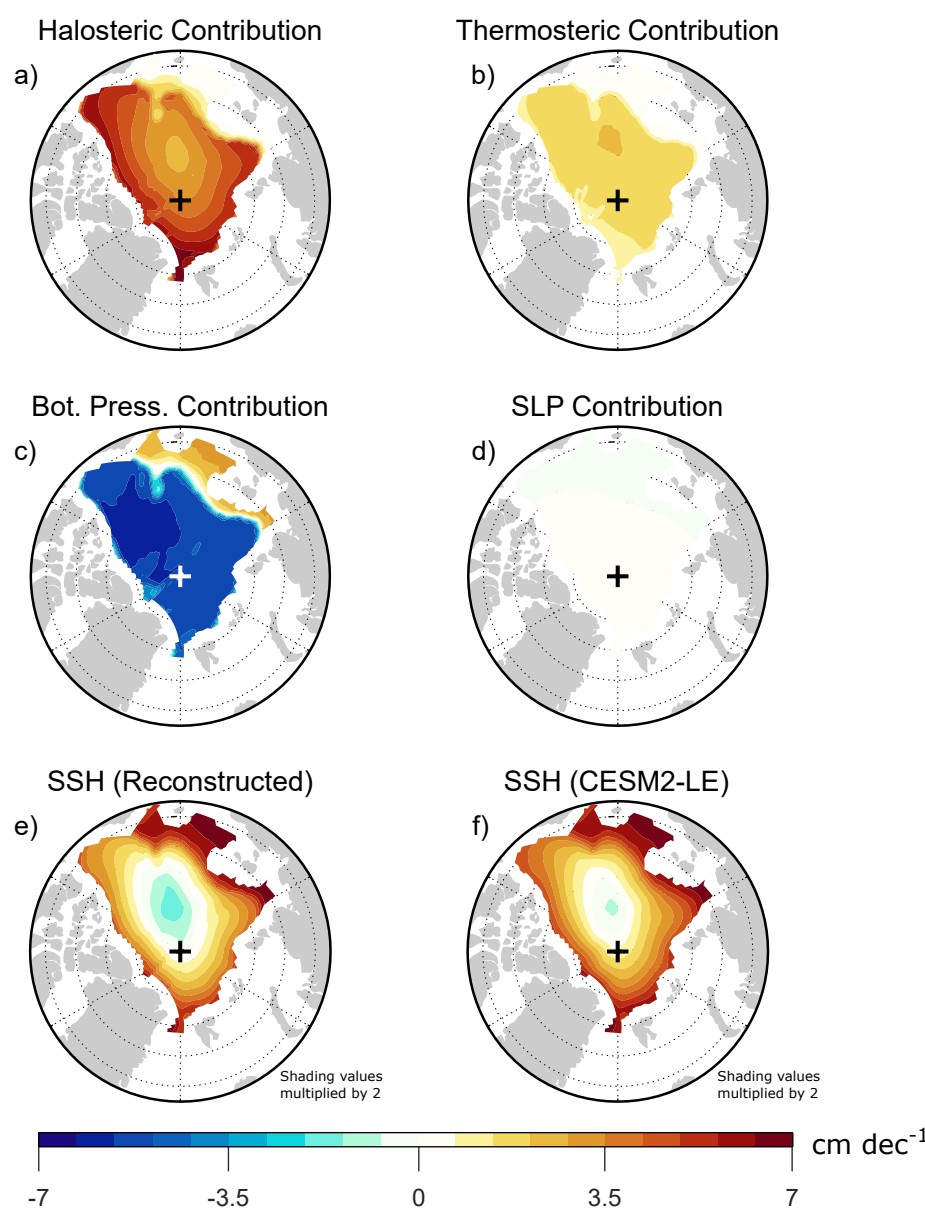

**Figure 8.** CESM2-LE September 2025-2100 trends of (a) the halosteric contribution to SSH, (b) the thermosteric contribution to SSH, (c) the bottom pressure contribution to SSH, (d) the SLP contribution to SSH, (e) SSH reconstructed from the sum of panels a-d, and (f) total SSH directly from CESM2-LE output. Trends are computed from monthly mean timeseries as detailed in section 2. Note that, for clarity, the shading scale is different from that used in Fig. 7, and the shading values in panels e and f have been multiplied by two.

pressure decreases over the interior Arctic offset some of the steric SSH increase. We have confirmed that the contribution of SLP changes is negligible (Fig. 8d), as it is approximately an order of magnitude smaller than the other contributions. The SSH reconstructed from the sum of the halosteric, thermosteric, bottom pressure, and SLP contributions (Fig. 8e) is very close to the SSH trend directly from CESM2-LE output (Fig. 8f), with differences below 0.5 cm dec$^{-1}$ that are likely due to our choices of expansion coefficients. Altogether, our analysis shows that the SSH trend over most of the Arctic is due primarily to salinity decreases, enhanced by temperature increases and offset by bottom pressure decreases.

## 4 Discussion and conclusion

Our analysis shows that CMIP6 models and CESM2-LE produce AADS trends similar to CMIP5 models (Tandon et al., 2018). In climate warming scenarios, the projected AADS trends are generally positive during March and negative during September. The negative September trends are of particular interest because they contrast with the positive trends produced by buoy observations (Rampal et al., 2009; Tandon et al., 2018).

Using daily output from CESM2-LE, we showed that the negative September AADS trends are due to a combination of changes in wind stress and tilt forces. The wind stress changes correspond with an SLP decrease north of Greenland, which is expected with the northward shift of the jet streams (e.g., Yin, 2005; Moore et al., 2018). As for the changes in tilt forces, CESM2-LE projects SSH to increase over most of the Arctic Ocean, with greater increases on the basin periphery. These SSH changes produce geostrophic changes in sea ice drift that contribute to the slowdown of the Transpolar Drift and Beaufort Gyre. We gained some confidence in the realism of the CESM2-LE simulations because the projected positive trends in March AADS show contributions from internal stress and wind stress changes that are supported by earlier observational studies (Spreen et al., 2011), although a precise regional comparison is complicated by internal variability in historical trends.

Tandon et al. (2018) also found that, as the Arctic transitioned from complete to partial sea ice cover, simulated sea ice drift speed started to decrease, and they referred to this phenomenon as a "sea ice extent effect." Our analysis reveals that this transition arises because, when the sea ice is sufficiently thin, it approaches a free drift state in which internal stresses play a negligible role. This state contrasts with the force balance for an ice-covered Arctic, in which forces generated from internal stresses dominate over other forces, causing sea ice to move faster as it thins.

The CESM2-LE September AADS trend during the historical period is essentially flat, which disagrees with the positive trend in observations (Tandon et al., 2018). Thus, if CESM2-LE's SSH and wind stress mechanisms are at work in the real world, it is possible that internal stresses are stronger in the real world than they are in CESM2-LE. It is also possible that summertime changes in internal stress and wind-ocean stress do not offset each other as much in the real world as they appear to in CESM2-LE. Additional work is needed to compare the balance of these forces in models and observations.

Additional analysis of CESM2-LE output reveals that the positive summertime SSH trends are due primarily to freshening of the interior Arctic basin, with warming of the Arctic Ocean acting as a secondary contribution. The corresponding dynamical changes also lead to anomalous drift toward the Russian shelf region, resulting in a piling up of water that further reinforces the SSH increases. However, CESM2 (like other models) does not produce observed Atlantification in the Eurasian basin

(Muilwijk et al., 2023), and improved representation of this process might result in different SSH trends in the Eurasian basin and accordingly different sea ice motion trends.

Because of the unavailability of the required daily output, we were not able to confidently assess the relative contributions of various forces to projected sea ice motion changes in other models. But even with daily data, attribution of mechanisms is challenging because of strong subdaily variability during summer. Greater availability of subdaily data would allow a more rigorous quantitative attribution of mechanisms. Once such mechanisms are clarified across more models, progress can be made toward improving the models collectively and resolving disagreements with observations.

*Code and data availability.* CMIP6 data is distributed by the Earth System Grid Federation (ESGF) and can be found at https://esgf-node. llnl.gov/projects/cmip6 as described in Eyring et al. (2016). CESM2-LE data is maintained by the University Corporation for Atmospheric Research (UCAR) and can be accessed at https://www.cesm.ucar.edu/community-projects/lens2/data-sets as described in Rodgers et al. (2021). The code used for analyzing these data can be obtained from the authors upon request.

## Appendix A

This appendix contains additional figures that support specific points made in the text but are beyond the main focus of this study.

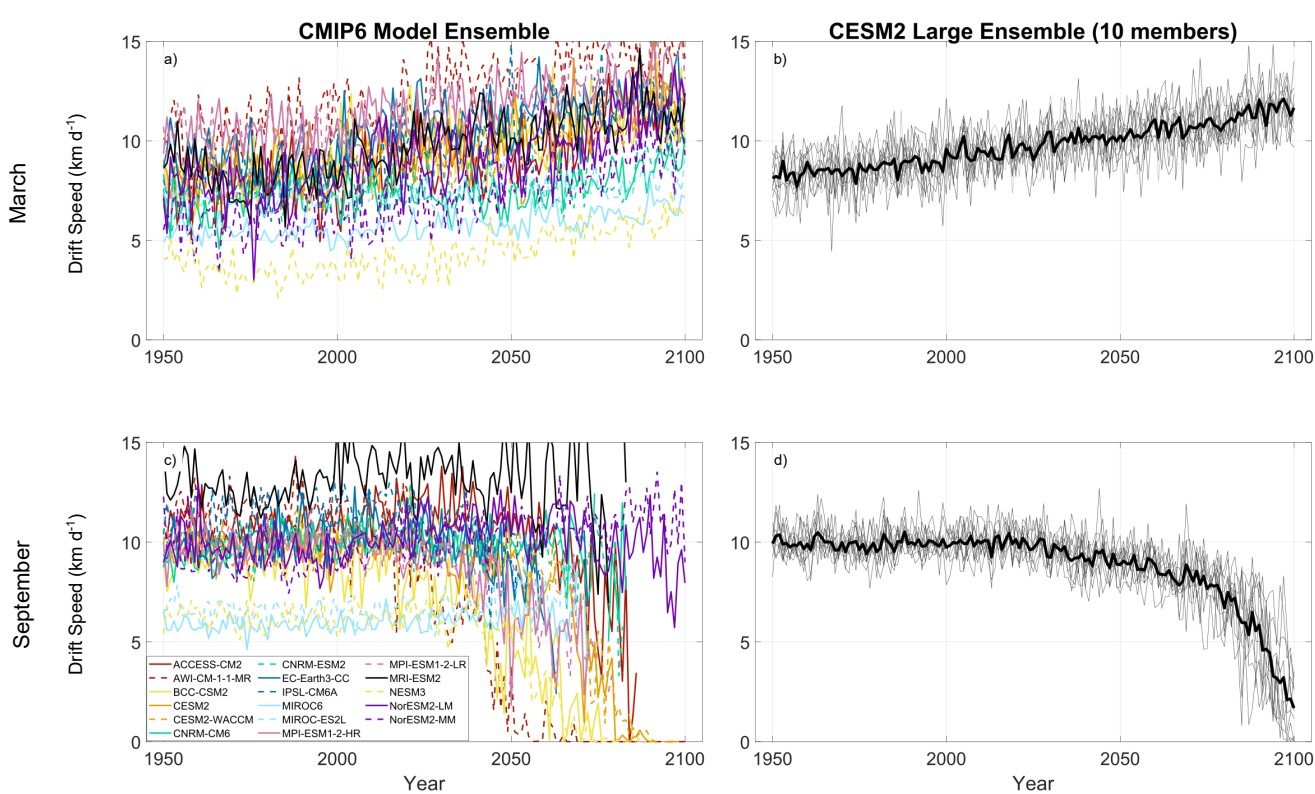

**Figure A1.** As in Fig. 1 but without temporal smoothing applied.

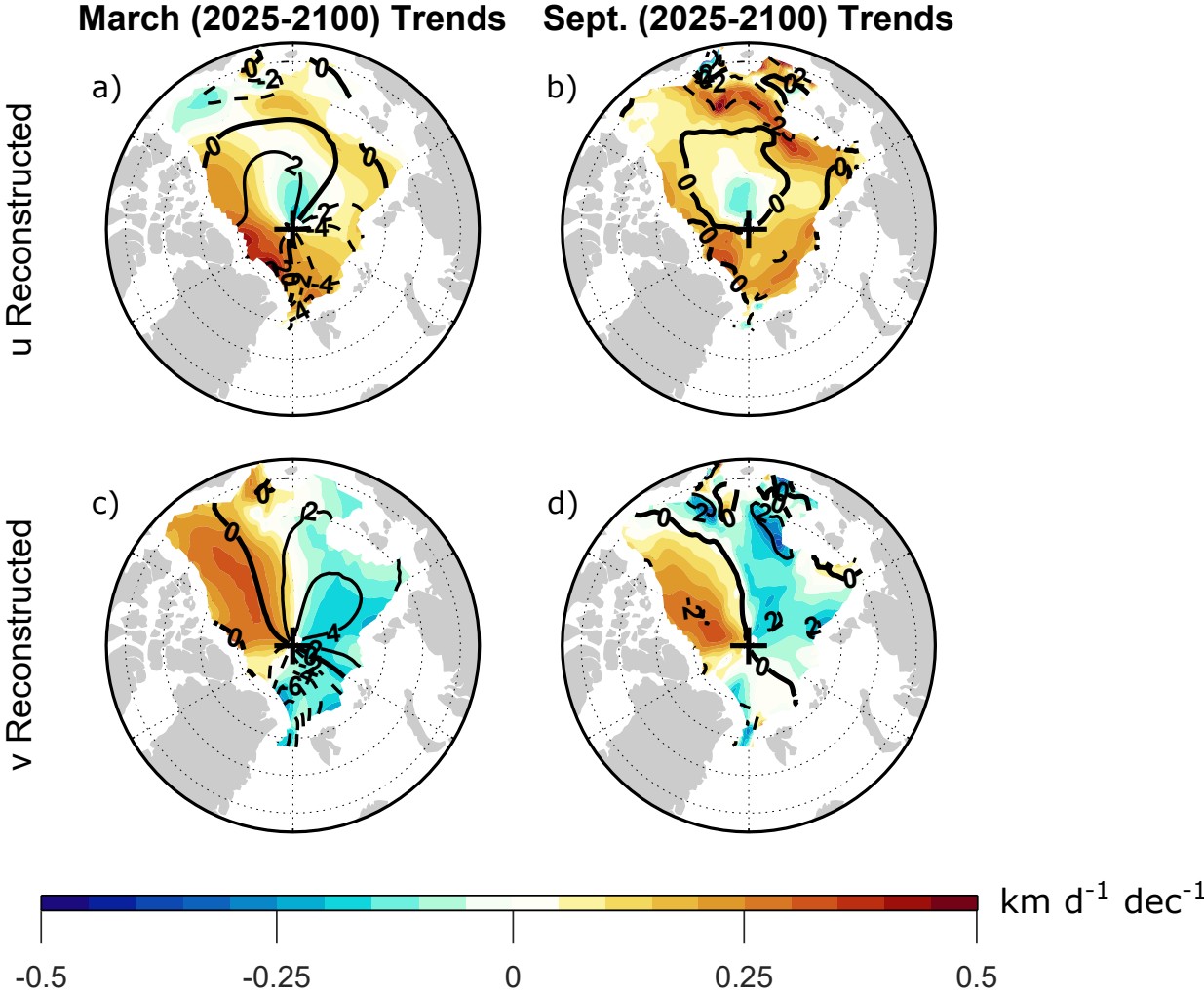

**Figure A2.** Shading shows the CESM2-LE trends obtained by summing the trends of the individual terms on the right hand side of the steady-state (a,b) zonal and (c,d) meridional momentum balance (equation 3) during (a,c) March and (b,d) September 2025-2100. Trends are computed from monthly mean timeseries as detailed in section 2. Contours show the climatological reconstructed velocity components averaged over 2015-2035, with contour interval of 2 km d$^{-1}$. Negative contours are dashed and the zero contours are thick.

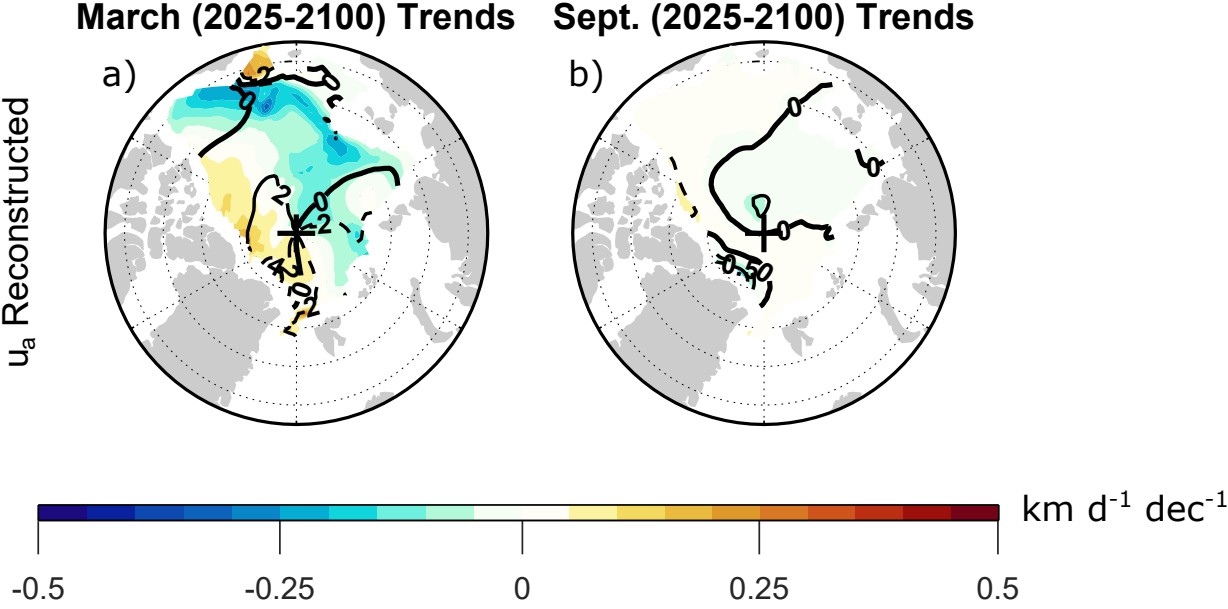

**Figure A3.** Shading shows the CESM2-LE trends obtained by summing the trends of the individual ageostrophic terms on the right hand side of the steady-state zonal momentum balance (equation 3) during (a) March and (b) September 2025-2100. Trends are computed from monthly mean timeseries as detailed in section 2. Contours show the climatological reconstructed velocity components averaged over 2015-2035, with contour interval of (a) 2 km d$^{-1}$ and (b) 0.5 km d$^{-1}$. Negative contours are dashed and the zero contours are thick.

*Author contributions.* JLW acquired the data, analyzed it, created manuscript figures and wrote the manuscript. NFT conceptualized the project, provided guidance for completion and revised the manuscript.

*Competing interests.* The authors declare that they have no competing interests.

*Acknowledgements.* We thank David Docquier and Steve Howell for helpful discussions. Three anonymous referees provided very valuable and constructive feedback on the submitted manuscript. We acknowledge the modelling centres that contributed to CMIP6.

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
