# Peer review of "Why is Summertime Arctic Sea Ice Drift Speed Projected to Decrease?"

_The Cryosphere, 2023_

## Author Comment (AC1)

We sincerely thank all of the referees for taking the time to thoroughly read our manuscript and provide very valuable feedback. Below please find the referee comments reproduced in black along with our point-by-point responses in red.

**Responses to Anonymous Referee 1**

Paper Evaluation

This paper is clearly written. The methods are precisely defined, the equations and figures are integrated well, and I barely found a typo. Well done on that part – especially with the equations. I too often find it's either no equations to demonstrate the physical theory or inadequate text provided to explain what the equations show. This paper has a good balance.

By the science, again, this is a strong study. The narrative line from research question to research design to results to conclusions is logical and organized. The authors do a good job emphasizing the September slow-down as the key point of interest while also providing context. They also deliver on the promise of that mystery by providing robust physical explanations supported by a combination of evidence from their results and theory. There's nuance and plenty of limitations, but the authors describe those without making me feel too bogged down in the detail. Job well done. Great science. I think this should be published after some minor revisions.

Thank you very much for the positive feedback, and we greatly appreciate your suggestions, which we address in more detail below.

Line-by-line Comments

1. Line 25-26: This sentence sounds contradictory, starting with "Record-low SIE has been frequently observed since the mid-2010s" and ending with "no record recorded during this time period". If talking about September SIE, the latter statement is correct (2012 is still the lowest). However, the thrust of this paragraph seems to be emphasizing a negative trend, so I wonder if the authors actually meant to convey that SIE has frequently been below the 5th percentile of 1981-2010 average daily SIE, or something like that.

Thank you for pointing this out. We will replace "record recorded" with "record-high SIE observed," so the sentence will read "Record-low SIE has been frequently observed since the mid-2010s, with no record-high SIE observed during this time period."

2. Line 100: Sorry, this is a long comment for a small problem. It might even be minor enough to not be noticeable in the end, but because it might change a minor result noticeably, I must mention it.

The NESM3 model fields downloaded from the ESGF are still on the rotated ocean grid, so the smallest grid cells by area do not neatly align with the highest latitude. Therefore, a simple latitude-weighting will introduce some bias, providing too little weight approaching 90°N (e.g., rows 366 and 367, columns 159 and 160 in the NESM3 ocean grid, which all have a latitude of 89.7°N) and too much weight for certain other locations (e.g., latitude = 80.3°N, longitude = 320.4°N, which is row 383, column 159 in the NESM3 ocean grid). (Note: Those indices assume counting starts at 0.)

If you use the lat_bnds and the lon_bnds variables, you can find the latitude and longitude of the four corners of each cell and then calculate the area of the quadrilateral using line integrals (with Green's Theorem) or using Girard's Theorem. Line integrals is what MatLab's areaint function uses, and it seems to be preferred in GIS software (from what I can tell). But either works better than the simple cosine method for this application. There are example Python implementations of each here:
https://stackoverflow.com/questions/4681737/how-to-calculate-the-area-of-a-polygon-on-the-earths-surface-using-python.

If you're doubtful, let me attempt to convince you with an example. Let's compare two points I will arbitrarily call "point 1" (row = 337, col = 134) and "point 2" (row = 383, col = 159) in the NESM3 ocean grid.  The center of point 1 is (70.334°N, 105.192°E) and the center of point 2 is (80.313°N, 320.143°E). Based on the cosine of the latitude, the weight of point 1 should be 2.00 times larger than the weight of point 2. But point 2 is near the convergence point for the NESM3 ocean grid. So that's underestimating. Using Girard's Theorem and using the lat_bnds and lon_bnds variables, point 1 should really have 10.52 times more weight than point 2. Using Green's Theorem, point 1 should have 10.55 times more weight than point 2. That magnitude of difference was enough to convince me I should bring this up in the review. But again, if it only affects NESM3, there's no chance this has a major impact on any conclusions – only a minor impact for that one model in Figure 1 and A1.

p.s. I already had daily NESM3 sea ice fields on my computer and recently went through this entire process for a project of my own, so I was primed to be pedantic about this.

Thank you very much, this issue also occurred to us after submitting the paper. Fortunately, we were finally able to find areacello output for BCC-CSM2-MR and NESM3, so we will update the figures and text accordingly. As we would expect, this change has no impact on our conclusions. The updated version of Fig. 1 is shown below.

[Figure]

**Figure 1.** AADS simulated in (a,c) CMIP6 and (b,d) CESM2-LE during (a,b) March and (c,d) September. Each timeseries of each CMIP6 model and each CESM2-LE ensemble member is smoothed with a 21-year moving window. In panels b and d, the gray lines indicate individual ensemble members and the thick black lines indicate ensemble averages.

3. Line 117: I presume by "BCC-CSM2-1", the authors mean "BCC-CSM2-MR", as labeled in Table 1.

Yes, thank you, we will correct this.

4. Line 162: Replace "and much our analysis" with "and much of our analysis".

Yes, thank you, we will correct this.

5. Figure 1: The labels "March" and "September" need to be exchanged in this figure to match the caption and the text.

Yes, thank you, we will correct this.

6. Line 227-228: At first glance, it seems contradictory to declare that one can assume steady state for a variable undergoing a long-term trend. The resolution of that contradiction, of course, is the time scale, but it might be nice for the authors to note that explicitly. In other words, the daily ∂v/∂t can be ignored.

Thank you for pointing out this possible point of confusion. We will edit the sentence to read as follows: "Since we are interested in long-term averages and trends, we can assume that velocity

variations on this timescale are slow enough that the time tendency term ($\partial v/\partial t$) can be neglected."

7. Lines 312-318 or Lines 337-345: I can't find a paper that shows maps of Arctic summer precipitation or P-E change in CESM2 or CESM2-LE, but I do know that CMIP6 models in general produce positive trends in JJA precipitation over the Arctic under warming scenarios (McCrystall et al., 2021; IPCC AR6 WGI Figure 8.14), and the CESM2 produces positive trends in precipitation annually over the Arctic (Meehl et al., 2020). These ideas might be worth citing in discussion of the trend in sea surface salinity. The authors do a great job describing why the surface salinity pattern is different from the integrated salinity impact on sea-surface height, but I feel like there's a good opportunity to also discuss whether precipitation (directly or via river input) has any role to play in freshening at the surface (in like a sentence).

Thanks, this is a great suggestion. We will modify the text around lines 312-318 as follows: "Sea surface salinity (SSS) trends (Figs. 7d) are negative north of Greenland and Russia. The spatial structure of these SSS trends are similar to those found in the annual mean for CMIP5 and CMIP6 models, and they show a strong correspondence with river runoff increases (Wang et al., 2022). These SSS decreases would also act to reduce water density and contribute to increased SSH, while the positive SSS trends elsewhere would act to reduce SSH. These spatially varying SSS trends might be combining with the more spatially uniform SST warming to produce the spatially varying SSH trends. These results provide additional evidence that SSH trends are due to steric changes."

8. Figure 9: This is the only figure where I have trouble seeing everything the authors are describing. The SSH filled contours are clear, but the sea ice motion vectors are sub-optimal because there aren't enough of them for me to adequately see the spatial patterns. Of course, I can't just recommend adding more arrows because that risks over-crowding. Therefore, I also recommend the authors do any/everything they can to increase the map size, giving more space for arrows. For example, make the figure taller, make the quiver keys smaller (e.g., by declaring the units just once in the figure), and maybe even shift that color bar into the blank space in the lower-right.

If the authors don't think they can adequately plot more vectors, then I advise they make another figure (main or appendix) that shows maps of the September 1979-2014 SSH and motion climatology for each model. That at least would give a reference for readers when trying to visualize the description in the results. Actually, this might be a good idea even if the authors can improve Figure 9, but I only think it's a must-have if they the authors prefer to leave Figure 9 as-is.

Thank you for this feedback and the helpful suggestions. We have updated this figure as shown below to make the vectors easier to see.

[Figure]

**Figure 9.** CMIP6 projected September trends of SSH (shading) and sea ice drift velocity (vectors). For the model indicated above each panel, the trend is computed during the period of decreasing September AADS trends (the year ranges indicated in the last column of Table 1), excluding any years before 2025. NorESM2-MM is excluded from this figure because it does not produce negative September AADS trends. MIROC-ES2L and MIROC6 are excluded because the sea ice extent in these models is too low to produce a useful visualization of regional trends. For clarity, the shading values in panel c have been multiplied by four, and the shading values in panel g have been divided by two. The shading scale is different from that used in Fig. 7a.

**Responses to Anonymous Referee 2**

The authors present diagnostics of the sea ice momentum balances in a series of GCMs with the goal of understanding the projected decrease in sea ice drift velocities in the Arctic under a future increasing $CO_2$ scenario. It is found that changes in SSH in the models are primarily responsible for the change in drift velocity. While this is correct, I think it would be clearer to state that the change in drift velocity results from a change in the ocean currents. Since the wind stress and internal stresses do not change significantly that means the ice drift relative to the ocean also does not change (assuming the drag coefficient is similar), so it is really the ocean

currents that drive the change. This is of course directly related to SSH through geostrophy but I think the interpretation is clearer. In fact, if one assumes that the surface velocity in the ocean is in geostrophic balance with SSH gradients, the tilt terms drop out and the Coriolis term is acting on the difference between the ice and ocean velocities, making the underlying mechanism clear.

If the surface velocity of the ocean is in geostrophic balance with SSH gradients, that by definition means that the velocity in the Coriolis term is being precisely balanced by the SSH gradients. In this case, both the Coriolis and tilt terms drop out of the momentum balance, and in a steady-state, the atmospheric, ocean and internal stresses balance each other. The referee appears to argue that, under geostrophic balance, the tilt term drops out of the momentum equation but the Coriolis term doesn't drop out, which is not correct.

We think it is physically more accurate to state that the direct influence of SSH on sea ice motion is simply due to gravitational acceleration down the SSH slope. The SSH can in turn be influenced by a combination of steric changes and changes in ocean dynamics, but our analysis later in the paper shows that we cannot rule out the influence of river runoff on SSH. We acknowledge that ocean dynamical changes (e.g. weakening of the AMOC) are likely influencing the SSH changes, but we do not think that there is adequate evidence to claim that the SSH changes are entirely driven by ocean dynamics.

Having said all of that, there are reasons to believe that changes in atmospheric winds are playing a stronger role than we suggested in the submitted paper. This is because, as we detail in our response to referee 3 below, the mismatch between the total and reconstructed u trends implies a significant role for nonlinear ageostrophic processes. There is qualitative correspondence between this ageostrophic trend and changes in wind stress, suggesting a significant role for wind stress changes.

line 13: But the models do not reproduce the observed ADDS increase in summer for the historical period, so I think it is optimistic to assume that the models will be correct in the future. More explicit discussion of the model shortcomings in summer are needed (around line 64).

We do not claim that the models' summertime projections are correct. We state that the mechanisms at work in the models are likely also at work in the real world, but (as we suggest in the next sentence) there is reason to believe that the precise strengths of those mechanisms are different in the real world compared to the models. To further emphasize this point and avoid misinterpretation, we will insert text in the following sentence so that it will read "However, the precise strength of these mechanisms are likely not realistic during summer, and additional research is needed to assess whether the simulated summertime internal stresses are too weak compared to the tilt forces."

The paragraph around line 64 acknowledges the shortcomings of models in producing historical trends of summertime drift speed. Specifically, in lines 65-66 we state "CMIP5 models still underestimate observed summertime AADS increase over 1979-2014 (Tandon et al., 2018)." We are not aware of any earlier studies that have shown the reason for this underestimation, so at this point, we don't think it would be helpful to speculate about which model shortcomings are responsible for this underestimation. Thus, we will insert after this sentence "for reasons that are unclear" to convey that we are not leaving out any important discussion of earlier work. However, later in the paper (lines 391-395), we do indicate the relevance of our findings for understanding simulated historical trends, specifically that the underestimation of historical trends may be due to the contribution from internal stresses being too weak compared to the contribution from tilt forces.

line 26: no record recorded? Please clarify

Sorry for the typo here. We will replace "record recorded" with "record-high SIE observed," so the sentence will read "Record-low SIE has been frequently observed since the mid-2010s, with no record-high SIE observed during this time period."

line 67: What is the Representative Concentration Pathway? What is 8.5 W/m^2? Anomalous radiation averaged over the globe? Uniformly distributed? Same for line 83. I see this is explained layer, but maybe note that here.

Thank you, we will modify the text here as follows: "Tandon et al. (2018) also found a strong seasonal contrast in projected AADS trends under the Representative Concentration Pathway 8.5 (RCP8.5) scenario, in which globally averaged radiative forcing at the top of the atmosphere increases by 8.5 W m$^{-2}$ at the end of the 21st century. Under this scenario, March AADS steadily increases until the late 21st century for most CMIP5 GCMs, while September AADS trends switch from positive to negative in the early- to mid-21st century.

line 85: What does r1i1p1f1 (and similar) mean?

Here the numbers next to "r", "i", "p" and "f" provide labels for the realization (i.e. the ensemble member), initialization method, physics package, and forcing datasets, respectively. We will insert text to this effect.

line 101: Clarify - does this weighted averaging reflects the area of each model grid cell?

We found areacello for NESM3 and BCC-CSM2-MR, so we will delete this text and update the corresponding information in Table 1 and Figures 1 and A1. As we indicated in response to referee 1, these updates have no effect on our conclusions.

Line 105: What do you do in regions where the summer ice disappears over the duration of the experiment? If you neglect those points, how does this bias your estimate?

Thanks for the question. In this study, we are examining motion of sea ice, so it would be natural to discard points where there is no sea ice, and that is what we do. We will insert text here to make that clear. This discarding of points on its own does not introduce a bias, but if there is a bias in the sea ice cover, that could produce a bias in the sea ice motion calculation.

It should also be noted that if we didn't discard points where sea ice is absent, that would produce major impediments to the physical interpretation of sea ice motion changes, as sea ice motion declining to zero could be due to just the disappearance of sea ice rather than an actual decline in sea ice motion. By discarding points where sea ice is absent, as we have done, this ambiguity is removed.

Figure 1: It would be helpful if you could add the drift speeds derived from satellite data when it is available. This would help the reader to understand the shortcomings in the models representing the recent observational record. Also, it looks like the y-axis labels are incorrect.

Thanks, we will correct the y axis labels. We think that earlier studies like Tandon et al. (2018) have provided a satisfactory comparison of models and observations, and we have referred readers to this study. Since our study is focused on model projections rather than the historical period, and there is no major difference in overall model behaviour between CMIP5 and CMIP6,

we think it is sufficient to refer readers to Tandon et al. (2018), and introducing observational analysis in the current study would dilute the current study's focus.

line 171: Why is it supposed that the ice drift velocity results from a change in ice thickness and not a change in winds or ice-ocean drag coefficient (because ice is younger)?

Thanks for this feedback. Our statement here was meant to convey expectations based on earlier studies. But we agree that our analysis later in the paper shows that changes in winds are also influencing the projected drift speed changes in the western Arctic, and we will insert text here to anticipate that analysis.

Fig. 2c, d: It would be clearer to plot the change in drift as vectors. I am having a hard time visualizing the change from the individual velocity components. The same for Fig. 3. This would also reduce the number of panels. I do not see the benefit added for Figs. 4 c-f. All this information is contained in Figs. 4a and b.

Thank you for this feedback, and we gave a lot of thought to different ways of presenting our results. We include both velocity components and vectors in Figures 2, 3 and 4, and we think both are helpful. The individual components are helpful when decomposing into different contributions (i.e. tilt force, internal stress, etc.), as it is much easier to compare shading values for individual components rather than comparing vector angles and lengths. Furthermore, by plotting individual components, it is much easier to visualize the relationship to the climatology, since the change can be plotted as shading and the climatology can be overlaid as contours. Such visualization would be very difficult with vectors because it would require overlaying anomaly vectors on top of climatology vectors, and the plots would become overly busy.

To facilitate interpretation of the zonal component, we will add text that positive (negative) values indicate counterclockwise (clockwise) motion around the North Pole. To facilitate interpretation of the meridional component, we will add text that positive (negative) values indicate motion toward (away from) the North Pole.

line 237: Are the only assumptions that the ice balance is steady and linear?

We have assumed that the ice balance is steady and momentum advection is negligible. We have not assumed linearity, as temporal variations in sea ice mass per unit area ($m_A$) can produce nonlinearity.

line 254: Again, I think trend vectors would be clearer.

Readers wishing to see the vector velocity can refer back to Figure 4. The main purpose of Figure 5 is to assess the geostrophic and ageostropihic contributions to the velocity. As we stated in response to the referee's earlier comment, such comparison is much easier using a single component than using vectors, and shading allows for much clearer visualization of the climatology. As we stated above, we will add text to facilitate interpretation of the zonal component.

line 267: The lack of seasonality indicates that the geostrophic trends are not due to seasonal ice melt but instead probably due to changes in the permanent halocline.

Indeed, we think your suspicion is confirmed by our analysis of halosteric changes later in the paper. We will add text here to anticipate this point, so the sentence will read "Intriguingly, the

geostrophic trends during March and September are nearly identical (Fig. 5c,d), suggesting that processes other than sea ice melt are responsible for generating the sea surface tilt. (We will revisit this matter below when analyzing SSH changes.)"

Fig. 6: Can you interpret why the internal stress term changes in the way that it does? Is it due to convergence or shear?

As we state on lines 279-280, the internal stress changes are qualitatively opposite to the wind-ocean stress changes, indicating that the internal stress is reacting to the wind-ocean stress. If there were only normal stresses and no shear stresses in the sea ice, such a change could be understood much like a spring's internal force reacting in the opposite direction to an applied force. However, in reality and in models, there are also shear stresses in the sea ice and more work is needed to assess the possible contribution of shear stresses. We will add text to this effect.

Fig. 9: vectors are too small

Thank you for this feedback. In response to this feedback and feedback from other referees, we have updated this figure to make the vectors clearer. Please see our response to referee 1.

Discussion and Conclusion: The decline appears to be driven by a (spatially variable) freshening of the Arctic. However, in recent decades the Arctic below the halocline has been getting saltier as more Atlantic-origin waters are penetrating into the basin. Do the climate models reproduce this effect? If not, why should we believe what the climate models predict for the future? It should be discussed how well the models represent this important shift in the hydrography of the Arctic. Again, I think it would be clearer to state that the change in drift velocity is due to a change in ocean velocity rather than a change in SSH tilt.

As we stated in response to the referee's earlier comment, we do not agree that it would clarify matters to state that the projected changes in sea ice motion are due to changes in ocean velocity rather than SSH tilt.

We thank the referee for raising the important point of discussion regarding penetration of Atlantic-origin waters. Indeed, most models (including CESM2) do not reproduce the observed "Atlantification" in the Eurasian Basin (Muilwijk et al., 2023, doi:10.1175/JCLI-D-22-0349.1). However, Atlantification has not extended into the Amerasian basin, and models show increases in stratification there, in agreement with observations (Muilwijk et al., 2023). Nonetheless, depending on how this Atlantification competes with surface freshening due to increased precipitation and river runoff, there may be reductions rather than increases in SSH in the Eurasian Basin, which would affect changes in sea ice motion. Thus, further work is needed to improve the model representation of Atlantification in the Eurasian Basin in order to improve confidence in model projections of sea ice motion, and we will add text to this effect.

All figures are too small. Show only down to 70N would increase the area of interest.

Thank you for your suggestion. We will make this domain change in each figure. This adjustment was especially important for Fig. 9, and you can see an updated version of that figure in our response to referee 1.

**Responses to Anonymous Referee 3**

This paper aims to answer the question posed in its title: why do climate projections show a decrease in Arctic sea-ice drift speed? The authors analyse the outputs from 17 CMIP6 models and ten members of the CESM2 large ensemble. By decomposing the different contributions to the sea-ice drift in these models, they conclude that in winter, the drift increases because the internal ice stress decreases due to the ice thinning. In summer, they conclude that the drift decreases because of a reduced sea-surface height gradient caused by the freshening of the Arctic Ocean. The paper is clearly written, understandable, and logically structured. The paper's conclusions also appear well-founded and reasonable, but the attribution to SSH gradient in September needs clarification and further work - as discussed below.

I'm asking for major revisions because I want to review the revised manuscript. I'm not sure that there is a very substantial amount of work required, though.

We thank the referee for their valuable feedback, and we address their comments in more detail below.

I have two substantial comments and a few minor comments, as follows.

Firstly, in **line 92**, you say that "calculating drift speed from monthly output of drift components produces highly inaccurate results". This statement is not correct. Sea-ice drift speed is highly dependent on the time scale at which it is observed; i.e. calculating the drift speed from a daily displacement of a buoy and then taking a monthly average will give a very different result from calculating the speed directly from the monthly displacement. The same goes for using a model's monthly or daily velocity components to calculate the speed (as pointed out by Tandon et al., 2018). It is, however, important to note that both approaches are equally "correct" and "accurate". They are separate ways of observing the system from which we can learn different things. In this context, the only incorrect thing to do is to compare the speed obtained at a given time scale with that obtained at another - as Rampal et al. (2009) did.

Thank you, we agree with you. We will edit the text here as follows: "As discussed in Tandon et al. (2018), this time resolution is needed when comparing sea ice drift speed in models to drift speed from daily buoy observations."

Secondly, I struggled with the two paragraphs starting at line 237. In the first paragraph, you describe how you calculate the "reconstructed trends" and then say that these only capture 50% of the total velocity trend. We need a justification for the rest of the analysis when half the signal is missing, and this should come in the following paragraph, but I don't find it convincing.

You say that you tried using equation (3) to calculate the change in velocity between two days but that you got quantitative discrepancies. This conclusion doesn't make sense because equation (3) is a steady-state equation - so I assume you mean that you tried recreating the velocity field directly, not calculating the change.

Sorry for the confusion here. What we were trying to convey here (as you inferred) was that we attempted to recreate field the velocity field using the *transient* momentum equation, i.e. equation 2, not equation 3. (We should have indicated here the specific equation number, and we will do so in the updated manuscript.) If subdaily variability were insignificant, one should be able to numerically integrate equation (2) with respect to time using daily output on a given day and reproduce the velocity on the next day. In other words, given the SSH and sea ice fields on

day 1, one should be able to compute the sea ice velocity on day 2 from equation 2 if subdaily variability is insignificant.

During September, we were not able to accurately compute sea ice velocity this way, which suggests that subdaily variability of sea ice is significant during September. In contrast, we were able to compute sea ice velocity this way during March with reasonable accuracy, suggesting that subdaily variability is much less significant during March than during September, and this agrees with expectations based on the earlier studies we discuss in the paper. Overall, this analysis strongly suggests that subdaily variability is much more significant during September than during March, and thus we expect that any momentum budget calculation using daily output would be much more inaccurate during September than during March. We will update the text to make these points clearer.

Still, you ascribe the discrepancies between your results and the model fields to sub-daily variability, and I also have a problem with this. If we can assume a steady state (as is appropriate for climate models), then equation (3) holds, and as it is linear, then the mean equals the mean of the components. Sub-daily variability, while present, doesn't enter into it. It is even questionable how much sub-daily variability is present in a climate model - but this is a different story. So, I'm not convinced that you miss 50% of the trend to sub-daily variability, but even if you do, that begs the question of why the sub-daily variability decreases, which you need to address.

If there were no variability in sea ice mass per unit area ($m_A$), then equation 3 would indeed be linear, and subdaily variability would indeed be irrelevant. But if $m_A$ does vary on subdaily timescales, then equation 3 becomes nonlinear. Like the referee, we were skeptical that subdaily variability was the issue, which is why we performed the additional tests described in the response to the previous comment.

Having given it further thought, since the SSH term in equation (3) is linear, then the only source of inaccuracy in the reconstructed trends would be trends in the nonlinear ageostrophic terms. Thus, we can be confident that our calculation of the geostrophic velocity trend is accurate, and a more accurate way of calculating the ageostrophic contribution is by subtracting the geostrophic trend from the total velocity trend. Below we show an updated version of Fig. 5 in which the ageostrophic velocity trend is computed this way.

[Figure]

**Figure 5.** Shown in shading are the CESM2-LE trends computed over 2025-2100 for (a,b) the geostrophic zonal drift component, and (c,d) the ageostrophic zonal drift component during (a,c) March and (b,d) September. See the text for details on these computations. Contours show climatological values averaged over 2015-2035, with contour intervals of (a-c) 2 km d$^{-1}$ and (d) 1 km d$^{-1}$. Negative contours are dashed and the zero contours are thick.

Comparing the updated Fig. 5c with our original Fig. 5e shows that the two approaches produce nearly identical results during March, as we would expect since subdaily variations are not an issue during March. However, comparing the updated Fig. 5d with our original Fig. 5f shows that the two approaches produce very different results during September, as we would expect since there are much stronger nonlinear effects due to subdaily variability during September.

Furthermore, the ageostrophic changes in the updated Fig. 5d show a qualitative correspondence with the changes in wind stress shown in Fig. 6d and are qualitatively opposite to changes internal stress and ocean stress. This analysis suggests that changes in winds are also an important contribution to projected summertime decreases in sea ice motion, with changes in ocean and internal stresses acting as secondary modulating contributions. However, due to the unavailability of subdaily model output, more direct and quantitative confirmation of the role of wind stress changes has to be left for future work.

We are not sure that we understand the referee's last point "that begs the question of why the sub-daily variability decreases, which you need to address." We do not claim that sub-daily variability decreases over the long term during September. However, we do see evidence that subdaily variability is greater during September than during March and that claim is supported by earlier studies and the analysis we describe in our response to the previous comment.

Perhaps the referee's comment should be interpreted more generally to be asking "what happens with the other 50% of the reconstructed trend?" as in their comment further below, and our modified approach to calculating the ageostrophic velocity, as described above, addresses that question.

Ultimately, I don't think computing the ensemble mean, as you have done, is the right way to go. I assume you calculate each term of equation (3) for each member and then work with the mean of these across the ensemble at a daily time scale. If you do this, you are essentially filtering out the synoptic signal; each member will have different weather systems, so the mean will only give you the long-term motion. Comparing this against the speed calculated at the daily time scale is not appropriate, and I suspect this is why the reconstructed trends are 50% smaller than the actual trends. Doing the reconstruction with monthly values for the component terms would be more appropriate, as this also filters out the synoptic motion. There is probably a more sophisticated way to do this, but I can't think of one now.

Indeed, if we were comparing a long-term trend of daily velocity output to an ensemble average computed from the right hand side of equation 3, it would be problematic, but that is not what we did. Apparently it was not clear in our methodology, but our approach is accomplishing what the referee is suggesting, but with an important additional detail. For clarity, let's consider an expansion of the meridional component of equation (3):

$$u = -\frac{g}{f}\frac{\partial H}{\partial y} + \frac{\tau_{ay}}{fm_A} + \frac{\tau_{wy}}{fm_A} + \frac{F_{iy}}{fm_A} \qquad (3')$$

Then our trend calculations are performed in the following steps.

1. We compute each term in equation (3') on each day during September using daily model output.
2. We average the results of step 1 over each September month to produce a monthly timeseries for each term at each gridpoint. At a given gridpoint, any days on which sea ice is missing are excluded when calculating the monthly average.
3. We compute the linear trend for each term over the monthly timeseries at each gridpoint. At a given gridpoint, any months for which sea ice is missing on all days during that month are excluded from the trend calculation.
4. We repeat steps 1-3 for each ensemble member and average over the ensemble. The "reconstructed trend" is the sum of the ensemble mean trends for each term on the RHS of equation (3').

It is hopefully clear now that our method is accomplishing what the reviewer suggests, except that, before taking monthly averages, we first compute the terms using daily output. This step is important for terms involving $m_A$, since submonthly variations in $m_A$ and stresses can result in inaccuracy if one were to compute their products using monthly output. Furthermore, the terms on both the LHS and the RHS of (3') are being averaged in exactly the same way, so there is no inconsistency regarding timescale. We will modify the text in our methods and results sections to make these points clearer.

If I am right above, your conclusion holds, with the caveat that it only pertains to the long-term motion. This is perfectly reasonable, but you can't state this about the synoptic scale motion because your method loses you 50% of the trend. This is probably not a problem because the ice is essentially in free drift in September in future scenarios, and we don't expect a trend in

any of the terms - except SSH. The slow-down must, therefore, indeed be due to the reduction in the SSH slope. But the details are fuzzy as the paper stands.

To cut a long story short: I think you've reached the right conclusion, but you need to explain better what happens with 50% of the trend in your reconstruction and why that doesn't matter for the results.

Hopefully our responses to the above comments provide additional reassurance that the inaccuracy in the reconstructed trends is indeed due to subdaily variability rather than a problem with our ensemble averaging approach. It would be nice to be able to show direct evidence using subdaily output, but such output is not available. So more direct confirmation of the role of subdaily variability and a more accurate reconstructed trend calculation will have to wait for future studies, and we will add text to this effect.

Furthermore, we have given further thought to this matter, and as we explain above, subdaily variability can only be an issue for the ageostrophic terms in equation (3). Our additional analysis then suggests that changes in wind stress are likely also contributing to the negative drift speed trends during September, and we will update the manuscript to make these points.

Minor comments:

L42: You say that improved models produce results that better agree with observations, but this is not the point in Kay et al. (which is now published in JAMES, btw). They just tune some parameters to get better results.

Thank you, we will update the reference to Kay et al. (2022) and edit the text here to read as follows: "Recent model developments, such as parameterizing surface melt ponds (Flocco et al., 2012), adding in "mushy layer" thermodynamics (Bailey et al., 2020) to represent sea ice surface properties more comprehensively, and reducing sea ice surface melt (Kay et al., 2022) produce output that better agrees with sea ice observations."

L99: Skip the text in the parentheses and just say grid cell area instead of areacello variable in the line below.

For reproducibility, we think it is helpful to state the actual variable names used in the computations, so we will not remove "areacello" here.

L119: It's not really a "pole hole", but rather a coordinate singularity.

Thank you. We will replace "pole hole" with "coordinate singularity" here.

L125: Why are you using different scenarios? Isn't that a problem?

Thank you for raising this point. We expect that the choice of scenario will have an impact on the strength of the climate change signal compared to noise. Thus, we have chosen to use the SSP85 scenario for the CMIP6 models, so that the climate change signal in each of the individual realizations is maximized. SSP85 output was not available for CESM2-LE, but this is not as much of a concern since the large number of ensemble members allows noise from internal variability to be effectively filtered out. Comparing the CESM2-LE results in Fig. 1b,d to the CESM2-CMIP6 results in Fig. 1a,c, we see that positive March trends and the negative September trends are stronger in SSP85 compared to SSP370. Nonetheless, the trends are

qualitatively similar, and the CESM2-LE results fall within the range produced by CMIP6 despite the different scenarios. Thus we expect that the mechanisms of projected sea ice motion in CESM2 under SSP370 to be relevant for understanding mechanisms in CMIP6 models under SSP85. We will add text to this effect.

Figure 1: It would be nice to have the observed drift speed on these graphs as well. In the CESM2-LE graphs, the lines for the individual members are almost invisible. The legend is too small to read. The Y-axis labels are switched (September should be March, and vice versa).

Thank you, we have updated Fig. 1 in accordance with this feedback, and an updated version can be found in our response to referee 1.

Regarding observed drift speed, we refer to Tandon et al. (2018) who performs comparison with observations. Since our focus is on model projections and there is no major change from CMIP5 to CMIP6, we do not think it adds much to include observations here, and doing so risks diluting the focus of a paper that is already quite long.

I'm suspicious of the velocities going to zero in September. This probably coincides with the Arctic becoming ice-free in these models, but the velocity is undefined in that case - not zero. It looks like you made a mistake with the area averaging.

As we state in the methods, we discard any points where sea ice is missing, so sea ice velocities that go to zero are not due to averaging in ice-free areas. Zero sea ice velocity here indicates that the simulated sea ice velocity actually vanishes. There are other models, such as IPSL-CM6A, that become ice-free during September without AADS approaching zero, further reassuring that we have spatially averaged correctly.

Figures 2 - 9: Those figures have a lot of white in them. You include all of the Greenland, Norwegian, and Barents Seas, which are not of interest here. And there's a lot of space between subfigures. Reducing this would allow you to show more detail.

Thank you, we have adjusted the figure limits to show only latitudes north of 70°N (as suggested by referee 2) and reduced the amount whitespace between figure panels. These adjustments were especially important for Fig. 9, and you can see the updated version of Fig. 9 in our response to referee 1.

Figures 4-6: The contours are unclear and I'm not sure how useful this way of presenting the results is. But it could be good with larger figures.

Thank you. As we mentioned in response to the previous comment, we have adjusted the figure limits and the spacing between panels to reveal more detail. We include updated versions of Figs. 4 and 6 below. An updated version of Fig. 5 was provided above in response to an earlier comment.

Regarding the referee's comment "I'm not sure how useful this way of presenting the results is": Perhaps like referee 2, referee 3 is also questioning our choice to focus on velocity components rather than velocity vectors. If that is the case, we refer the referee to our response to referee 2.

Updated Fig. 4:

[Figure]

**Figure 4.** Shown in shading are the CESM2-LE trends computed over 2025-2100 during (a,c,e) March and (b,d,f) September for (a,b) sea ice drift speed trends (arrows depict sea ice drift velocity trends), (c,d) the eastward sea ice drift component, with positive (negative) values indicating anomalous counterclockwise (clockwise) motion around the North Pole, and (e,f) the northward sea ice drift component, with positive (negative) values indicating anomalous motion toward (away from) the North Pole. Contours show climatological values averaged over 2015-2035, with contour interval of 2 km d$^{-1}$. Negative contours are dashed and the zero contours are thick.

Updated Fig. 6:

[Figure]

**Figure 6.** Shown in shading are the (a,b) internal stress, (c,d) wind stress, (e,f) ocean stress, and (g,h) wind plus ocean stress contributions to the 2025-2100 trends of the zonal component of sea ice drift velocity in CESM2-LE during (a,c,e,g) March and (b,d,f,h) September. Contours show climatological values averaged over 2015-2035, with contour intervals of (a,g) 4 km d$^{-1}$, (b,h) 0.5 km d$^{-1}$, (c,e) 8 km d$^{-1}$, and (d,f) 2 km d$^{-1}$. Negative contours are dashed and the zero contours are thick. For clarity, the shading values in panels c-f have been divided by eight.